# ALCHEMY: AMPLIFYING THEOREM-PROVING CAPABILITY THROUGH SYMBOLIC MUTATION

**Shaonan Wu** [1,2,*]  **Shuai Lu** [3,†]  **Yeyun Gong** [3,]  **Nan Duan** [3,]  **Ping Wei** [1,2,†]

[1] National Key Laboratory of Human-Machine Hybrid Augmented Intelligence

[2] Institute of Artificial Intelligence and Robotics, Xi'an Jiaotong University

[3] Microsoft Research Asia

{shaonanwu@stu.,pingwei@}xjtu.edu.cn,
{shuailu,yegong,nanduan}@microsoft.com

## ABSTRACT

Formal proofs are challenging to write even for experienced experts. Recent progress in Neural Theorem Proving (NTP) shows promise in expediting this process. However, the formal corpora available on the Internet are limited compared to the general text, posing a significant data scarcity challenge for NTP. To address this issue, this work proposes *Alchemy*, a general framework for data synthesis that constructs formal theorems through symbolic mutation. Specifically, for each candidate theorem in Mathlib, we identify all invocable theorems that can be used to rewrite or apply to it. Subsequently, we mutate the candidate theorem by replacing the corresponding term in the statement with its equivalent form or antecedent. As a result, our method increases the number of theorems in Mathlib by an order of magnitude, from 110k to 6M. Furthermore, we perform continual pretraining and supervised finetuning on this augmented corpus for large language models. Experimental results demonstrate the effectiveness of our approach, achieving a 4.70% absolute performance improvement on *Leandojo* benchmark. Additionally, our approach achieves a 2.47% absolute performance gain on the out-of-distribution miniF2F benchmark based on the synthetic data. To provide further insights, we conduct a comprehensive analysis of synthetic data composition and the training paradigm, offering valuable guidance for developing a strong theorem prover. [1]

## 1 INTRODUCTION

Nowadays, some pioneer mathematicians are attempting to verify their proofs using the proof assistant Lean (de Moura et al., 2015; Tao, 2023). Writing proofs for formal statements demands mastery of formal language and domain-specific mathematical knowledge. To mitigate the complexity associated with completing proofs, several research efforts (Polu & Sutskever, 2020; Polu et al., 2023; Trinh et al., 2024) seek to automatically generate formalized proof through a neural model, known as Neural Theorem Proving (NTP). NTP represents a long-standing challenge for machine learning-based methods (Li et al., 2024), highlighting the limitations in the reasoning abilities of neural models. Prevalent Large Language Models (LLMs) (Brown et al., 2020; Dubey et al., 2024) still struggle with theorem-proving, despite excelling in related reasoning-intensive scenarios such as math reasoning (Reid et al., 2024) or code generation (Guo et al., 2024).

The key challenge of theorem-proving lies in data scarcity (Li et al., 2024; Trinh et al., 2024). Due to the difficulties associated with the manual formalization of theorems, formal corpora available on the Internet are relatively scarce compared to the general text (Azerbayev et al., 2024). Synthetic data has shown promise in alleviating the data scarcity problem. Some works propose to directly create theorems in symbolic space. For instance, Wang & Deng (2020) attempts to train a neural theorem generator on human-written formal theorems for the low-weighted formal system Metamath. Other efforts focus on generating theorems based on symbolic rules (Wu et al., 2021; Trinh et al.,

---

[*] Work done during internship at Microsoft Research Asia.

[†] corresponding author.

[1] The code is available at https://github.com/wclsn/Alchemy.

2024), which are restricted to a specific domain of mathematics, such as inequality theorems and 2D geometry. Additionally, there are endeavors focusing on autoformalization (Xin et al., 2024; Ying et al., 2024), which typically translates natural language mathematical problems into formalized statements, samples correct proofs, and retrains the theorem prover iteratively. Autoformalization has yielded promising results in competition-level theorem-proving tasks through the use of large autoformalized datasets (Xin et al., 2024). However, the process of formalizing problems and retrieving proofs is labor-intensive and cost-prohibitive. The distribution of formalized theorems is constrained by the pool of human-collected natural language problems and the intrinsic capabilities of the model. Compared to autoformalization, synthesizing theorems in symbolic space is a more direct process without intermediate translation, and is also easier to scale up to large, cost-effective CPU units.

Building upon the advanced Lean theorem prover (de Moura et al., 2015), we introduce a general method that synthesizes theorems directly in symbolic space. We analogize theorem synthesis to constructing functions in general programming language and adopt an up-to-down approach. Initially, a new statement (function declaration) is constructed for each candidate theorem. Specifically, with the mathematical library of Lean Mathlib[2] as seed data, we aim to find a symbolic manipulation between two existing statements. We posit that Lean's tactics serve as suitable candidates for manipulation because of their efficacy in handling symbolic expressions. $\{rw, apply\}$ are two basic tactics frequently used in theorem proving and capable of handling the equality and implication relationship between terms. We assign both tactics to the set of manipulations and retrieve the invocable theorems for each candidate theorem by executing a predefined list of instructions in an interactive Lean environment. Then we mutate the candidate statement by replacing its components with their corresponding equivalent forms or logical antecedents. Ultimately, we construct the corresponding proof (function body) based on the existing proof and verify its correctness using Lean. The worked example shown in Fig.1 illustrates the entire procedure of our algorithm. This algorithm is executed on a large CPU-only computing unit for several days. Our method increases the number of theorems in Mathlib by an order of magnitude from 110,657 to 6,326,649. This significant increase in the number of theorems demonstrates the potential of creating theorems in symbolic space.

We pre-train the LLMs on the combination of Mathlib theorems and their mutated variants. Then we fine-tune the models on the extracted state-tactic pairs, composing both the training split of Mathlib and additional synthesized state-tactic pairs. We demonstrate the effectiveness of our method by evaluating the theorem-proving capability of these provers on the challenging *Leandojo* benchmark (Yang et al., 2023). Our synthetic data improve the performance by 4.70% (over 70 theorems) on the novel_premises split. Furthermore, the synthesized data exhibit promise in enhancing the out-of-distribution theorem-proving ability of LLMs, as evidenced by a performance increase of about 2.47% on the competition-level miniF2F benchmark (Zheng et al., 2022).

Our main contributions are as follows. To the best of our knowledge, this work represents the first general data synthesis framework in the symbolic space for the Lean theorem prover, effectively complementing mainstream autoformalization-based methods. Notably, our synthesis pipeline increases the number of theorems in Mathlib by an order of magnitude. Associated code has been made open-source to facilitate further research in data synthesis for formal systems. Also, the synthesized theorems can serve as a valuable supplement to Mathlib. We conduct a comprehensive evaluation on both in-distribution and out-of-distribution benchmarks, providing empirical insights to enhance the theorem-proving capabilities of LLMs.

## 2 RELATED WORK

**Neural Theorem Proving**. Proof assistants such as Lean (de Moura et al., 2015), Isabelle (Paulson, 1994) or Coq (Barras et al., 1997) are gaining traction within the mathematical community. These tools help mathematicians in interactively formalizing and checking the correctness of proofs (Tao, 2024). Neural networks have shown promise in lowering the barrier of using a specific formal language for mathematicians, serving as a copilot (Song et al., 2024; Welleck & Saha, 2023). Polu & Sutskever (2020) propose to prove theorems automatically by training a decoder-only transformer to predict the next proofstep and construct the entire proof through a predefined search tragedy. Then a series of works seek to enhance the efficiency of this framework by incorporating auxiliary training

---

[2]https://github.com/leanprover-community/mathlib4

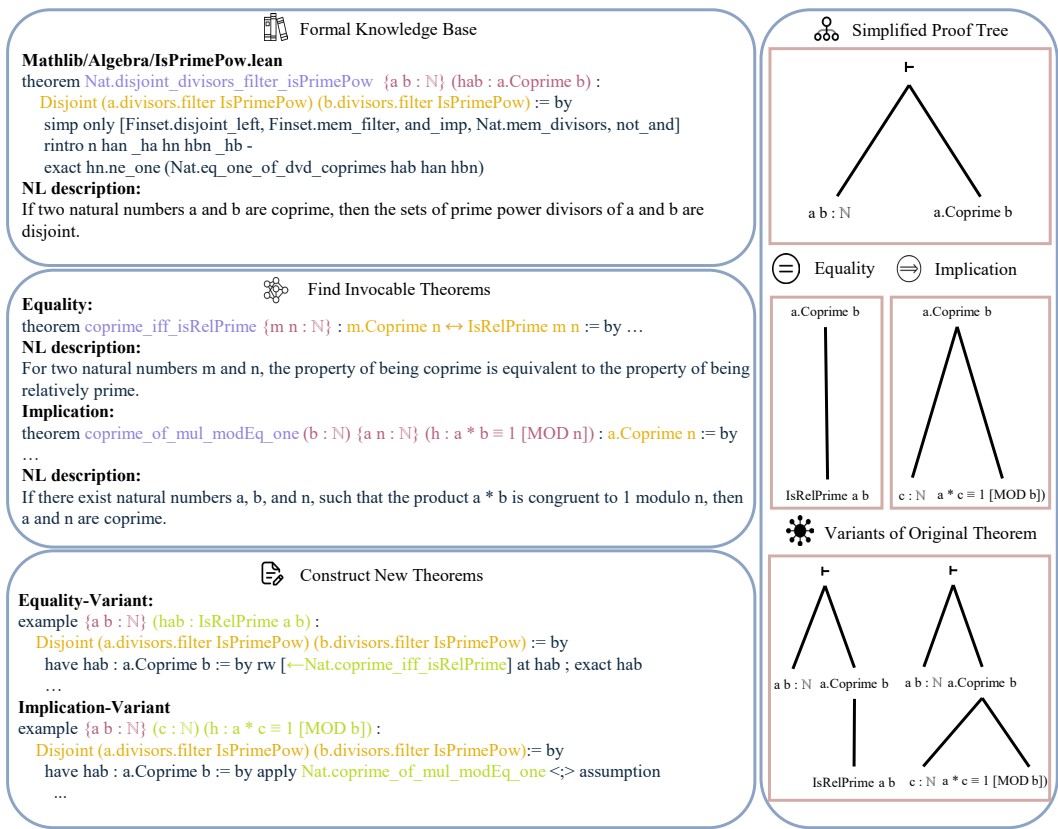

Figure 1: The overview of our synthesis pipeline. At the theorem level, we find invocable theorems that can be used to rewrite or apply to the assumptions or assertion of the candidate statement, such as the *iff* and implication rules about the *Coprime*. Then, we construct the new statements by replacing the specific component with its equivalent form or antecedent. At the proof tree level, our method merges two existing proof trees.

objectives (Han et al., 2022), conducting reinforcement learning (Polu et al., 2023; Xin et al., 2024), improving proof search tragedy (Lample et al., 2022; Wang et al., 2023; Xin et al., 2024), refining the premise-selection (Mikula et al., 2024; Yang et al., 2023) and so on.

**Synthetic Theorem Creation**. Data scarcity is a main challenge for NTP (Li et al., 2024). Synthetic data can effectively alleviate this problem alongside manual data collection (Wu et al., 2024). The current approach for synthesizing theorems diverges into two pathways. For autoformalization-based methods, the prevalent statement-level autoformalization is to translate a set of natural language problems into formal statements, followed by expert iteration to sample a collection of proofs for these statements (Wu et al., 2022; Xin et al., 2024; Ying et al., 2024). The proof-level autoformalization (Jiang et al., 2023; Huang et al., 2024) leverages LLM to generate a proof sketch, which is completed by symbolic engines such as Sledgehammer (Böhme & Nipkow, 2010). In contrast, the second pathway focuses on synthesizing theorems in formal space. Wang & Deng (2020) propose to train a neural theorem generator to synthesize theorems on a low-weight formal system, Metamath (Megill & Wheeler, 2019) which has only one tactic *substitute*. Wu et al. (2021) sequentially edits the seed expression according to a predefined set of axioms and an axiom order to create a new statement, concatenating the implications from all steps to build a complete proof. This method is used to create theorems on domains grounded in well-established axioms, such as inequality theorems and ring algebra (Polu & Sutskever, 2020). Beyond these works, AlphaGeometry (Trinh et al., 2024) can solve olympiad geometry without human demonstrations by constructing statements and proofs in symbolic space from scratch, using a carefully designed deduction engine and large-scale computing resources. Our method aims to directly synthesize theorems in symbolic space on the advanced Lean theorem prover, fully utilizing the power of computing.

**Benchmarks for Theorem Proving**. Most neural theorem provers based on Lean are primarily trained on Lean's mathematical library, Mathlib. It encompasses a broad spectrum of mathematical subjects (e.g., algebra and analysis), composed of over 110,000 theorems along with their respective axioms and definitions. Researchers test the capability of neural models to prove in-distribution theorems on a held-out set of Mathlib (Polu & Sutskever, 2020; Han et al., 2022; Polu et al., 2023). Yang et al. (2023) creates a challenging data split of Mathlib (*novel_premise* split) which requires testing proofs to use at least one premises not seen in the training stage and mitigates the over-estimated phenomena in the traditional setting of evaluation (*random* split). Another widely-used benchmark, miniF2F, (Zheng et al., 2022) is a cross-system benchmark and includes competition-level problems as well as IMO-level problems in the domain of algebra and number theory.

## 3 METHOD

Theorems written in Lean can be viewed as a special form of code, where declarations and function bodies possess precise mathematical meanings. The initial step in creating a new theorem involves formulating a theorem statement (function declaration) that defines the essence of the theorem. Then, one must verify its correctness by generating a proof block (function body) and submitting it to the proof assistant for validation. The resulting theorems that pass type checking can serve as supplementary data for training a neural theorem prover. Following Polu & Sutskever (2020), we use proofstep prediction as the training objective and best-first-search as the search tragedy.

### 3.1 STATEMENT GENERATION

**Find invocable theorems**. Constructing a new statement is the first step in creating a Lean theorem. The candidate theorem $t$ has a statement denoted as $s$. In the corresponding Lean repository, there exists $M$ potentially invocable theorems $T_{pinv} = \{t_j\}_{j=0}^{M-1}$. We assume that the challenge in creating a new theorem involves effectively leveraging the possibly invocable theorem $t_j$ to mutate the candidate statement $s$. This understanding arises from two perspectives. Each theorem in Lean can be represented in the form of a proof tree as presented in Fig.1. The leaf nodes represent the assumptions, and the root node signifies the assertion. At the tree level, the task of generating a new Lean theorem with existing theorems is equivalent to defining manipulations $\Phi$ that combine the proof trees of $t_j$ and $t$. To streamline this process, our focus is solely on establishing the connection between the root node of $t_j$ and the leaf node (or root node) of the candidate theorem $t$. From a mathematical standpoint, we can transform a target formula into an equal variant or break it down into multiple subformulas that suffice to prove the original formula, by employing the equality or "only if" relationship between formulas. The mathematical interconnections between formulas provide heuristic insights on how to mutate $s$ to create a new theorem. Similarly, we can substitute the terms in $s$ with their equivalent forms or logical antecedents. For instance, consider the statement $a + b > c + d, m > 0 \rightarrow m(a + b) > m(c + d)$ and the known theorems $a > b \iff e^a > e^b$ and $a > c, b > d \implies a+b > c+d$. From these, we can derive new theorems: $a+b > c+d, m > 0 \rightarrow e^{m(a+b)} > e^{m(c+d)}$, and $a > c, b > d, m > 0 \implies m(a+b) > m(c+d)$. In summary, identifying manipulations $\Phi$ that use $t_j$ to modify the assumptions or assertion of $s$ is the primary step in constructing new statements.

With their intrinsic mathematical meanings and proficiency in manipulating terms within Lean, tactics are promising candidates for the manipulations $\Phi$. Following the preceding discussion, we choose two frequently used basic tactics, *rw* and *apply* to formulate $\Phi$.

- **rw**. The "rewriting" tactic *rw* is mostly used to replace some terms in the target expression with their equivalent forms according to the given identity or *iff* (a.k.a., if and only if) rules[3]. In the presence of an identity $h : a = b$ or an *iff* rule $h : P \iff Q$, *rw [h]* substitutes all occurrences of term on the left side of equality in the proof goal with term on the right side. The direction of substitution can be reversed by adding a back arrow in the bracket (*rw [← h]*). The target of rewriting can also be changed using *at*, e.g. *rw [h] at $h_1$*, where $h_1$ is an arbitrary assumption of the current proof state.

---

[3]Strictly speaking, the *rw* tactic is used to handle equality in Lean. The identity and *iff* are just some kinds of equality.

Table 1: Templates for instructions designed to be executed in a Lean environment. We determine if a theorem is invocable by running the specific instruction.

| Tactic | Instruction Template | Description |
|--------|---------------------|-------------|
| | **Equality**   invocable_theorem : $a = b$ or $a \iff b$ | |
| *rw* | rw [invocable_theorem] | replace all $a$s in goal with $b$ |
| | rw [←invocable_theorem] | replace all $b$s in goal with $a$ |
| | rw [invocable_theorem] at assumption | replace all $a$s in assumption with $b$ |
| | rw [←invocable_theorem] at assumption | replace all $b$s in assumption with $a$ |
| | **Implication**   invocable_theorem : $a \implies b$ | |
| *apply* | have assumption := by apply invocable_theorem | set assumption as current proof goal, and try to argue backwards |

- **apply**. The *apply* tactic is a "suffice-to" tactic. Given an implication, it will match the consequent with the proof goal. If matched, it will transform the goal into the antecedent of the implication. With an implication rule $h : P \implies Q$ and a proof goal $Q$, then *apply [h]* will reduce the goal to proving $P$, which means that "proving P suffices to prove Q by implication". Similarly, *apply* can be used to modify the assumption by deducing the implication forward. With assumption $h_1 : P$, then *apply [h] at $h_1$* will change $h_1$ into $Q$, which means "If P is true, then we can assert Q is true by the implication".

---

**Algorithm 1** Find invocable theorems

---

**Input:** candidate statement $s$, potential invocable theorems $T_{pinv}$, instruction templates $I$
**Output:** invocable theorems $T_{inv}$     ▷ $T_{inv} : \{(init\_state, next\_state, instruction) \cdots\}$
$(env, init\_state) \leftarrow$ INIT$(s)$     ▷ initialize gym-like environment and retrieve initial state
$T_{inv} \leftarrow \emptyset$
**for** $t$ **in** $T_{pinv}$ **do**
    **for** $i$ **in** $I$ **do**     ▷ for each instruction template
        instruction $inst \leftarrow$ FORMAT$(t, i)$
        $next\_state \leftarrow$ RUN_TAC$(env, init\_state, inst)$     ▷ run a tactic specified by instruction $i$
and theorem $t$
        **if** VALID$(next\_state)$ **then**     ▷ if return a valid proof state
            Add $(init\_state, next\_state, inst)$ to $T_{inv}$
        **end if**
    **end for**
**end for**

---

To generate a new statement, we need to find the relationship between the candidate statement $s$ and the potentially invocable theorems $T_{pinv}$. The pseudocode outlined in Algorithm 1 describes the main procedure to find invocable theorems. The process involves initializing a gym-like environment to interact with Lean and extracting the initial proof state for the candidate statement. Then, the algorithm iteratively tests whether one theorem can be used to rewrite or apply to the candidate theorem leveraging the instruction templates shown in Table 1. Suppose the feedback from the interactive environment is deemed valid according to predefined criteria, the algorithm adds the proof states before and after the tactic running together with the respective instruction to the set of invocable theorems $T_{inv}$. More information about this process is described in Appendix C.2.

**Mutate statements**. After obtaining the initial set of invocable theorems, we applied some filtering rules to $T_{inv}$ to improve the quality of the data and lower the complexity of mutating statements. With filtered invocable theorems, we construct new statements by replacing the components with their equivalent forms or antecedents. Since we use tactics in Lean to formulate the manipulations $\Phi$, most symbolic manipulations are bypassed to the Lean proof assistant. What remains is just parsing and replacing. Specifically, for the candidate statement $s$ and instruction $i$, we utilize its abstract syntax tree to pinpoint the exact location within the code that requires modification. Then

we replace the corresponding parts with mutants parsing from the subsequent proof state generated by the execution of a specific tactic. The details of our algorithm are described in C.3.

## 3.2 PROOF GENERATION AND THEOREM VERIFICATION

Mutated statements can serve as useful lemmas for theorem-proving only if we can construct proofs that pass the verification of the proof assistant. We construct the entire proof using symbolic rules. Although neural provers or other automated theorem proving tools, such as hammer (Böhme & Nipkow, 2010)), can generate more diverse proofs than rule-based methods, they are compute-intensive and do not guarantee the correctness of the generated proofs. The idea of building a proof block is intuitive. Given that we only make a one-step modification to the statement, transforming the original proof state to a mutated proof state, a logical approach is to reverse the mutation and utilize the original proof to complete the remaining proving process. We use *have* tactic to restore the modified part of a statement (the original assumption or assertion) by introducing a lemma.

- **have**. The *have* tactic enables users to introduce new assumption into the current proof state if they can prove it. Given an assumption $h_1 : P$ and an implication rule $h_2 : P \implies Q$, a new assumption $h : Q$ can be added by *have h: Q := by apply $h_2$ at $h_1$; exact $h_1$*. This tactic is usually used to introduce helpful lemmas when proving a theorem.

In addition to its ability to introduce new assumptions into the proof state, *have* can be used in both tactic-style proof and term-style proof, which enlarges the margin for theorems to which our method can be applied. Apart from this, the additional *have* instruction transforms the mutated complex proof state into a canonical proof state. To some extent, this transformation is analogous to constructing an auxiliary point in geometry problems, which we assume will be beneficial for theorem proving in the general domain. Subsequently, we combine the original proof with this lemma to build the proof for the new statement. The details of the implementation of proof generation are depicted in the Appendix C.3. We construct the proof block for each mutated theorem. Then we submit the synthesized theorems to the Lean theorem prover for verification and remove the wrong ones. Details of the verification process are provided in Appendix C.4. Finally, we obtain $\hat{M}$ variants $V = \{v_j\}_{j=0}^{\hat{M}-1}$ defined by the keyword "example" for each candidate theorem.

## 3.3 MODEL TRAINING

Regarding the synthetic data, we have two observations. At the theorem level, the synthetic data comprises numerous theorems, each with statement distinct from existing theorems. At the state-tactic level, the process of constructing proofs introduces additional state-tactic pairs, primarily centered on *rw* and *apply*. Based on these insights, we assume that the synthetic data can serve as an augmented corpus for continual pretraining (CPT) and supervised finetuning (SFT). Specifically, we fine-tune LLMs using the proofstep prediction objective proposed by Polu & Sutskever (2020), utilizing state-tactic pairs derived from both seed theorems and synthetic theorems. Given the current proof state, the model is required to predict the next tactic sequence that contributes to the proving of the target theorem. We utilize the prompt template used by Welleck (2023), as shown in Fig.2.

```
/- You are proving a theorem in Lean 4.
You are given the following information:
- The current proof state, inside [STATE]...[/STATE]

Your task is to generate the next tactic in the proof. Put the next tactic inside [TAC]...[/TAC] -/
[STATE]
{state}
[/STATE]
[TAC]
```

Figure 2: Prompt template

## 4 EXPERIMENTS

We implement the data-synthesis pipeline described in Section 3 for *rw* and *apply*, constructing a set of variants for each candidate theorem in Mathlib. We train the LLMs on a mixture of human-written theorems and synthetic ones. To examine the effectiveness of synthetic data, we evaluate the theorem prover on two benchmarks that are widely adopted by the research community: 1) **Leandojo Benchmark** (Yang et al., 2023), which shares the same distributional characteristics as the seed theorems; 2) **miniF2F** (Zheng et al., 2022), a challenging benchmark focusing on competition-level problems that exhibits a distinct distribution compared to seed data. The experimental results derived from both benchmarks demonstrate the potential efficacy of our approach.

### 4.1 IMPLEMENTATION DETAILS

**Data-Synthesis**. We choose Mathlib4[4] which contains around 110k theorems as the seed data for data-synthesis. Our synthesis pipeline is built upon *Leandojo*[5] (Yang et al., 2023), a Python module that enables tracing a specific Lean repository, extracting the state-tactic pairs and abstract syntax trees (ASTs), and interacting with the Lean environment[6] (*run_tac API*). Finding invocable theorems is the most time-consuming step of our pipeline. For *rw*, the time overhead amounts to 14 days using 4,096 CPU cores[7]. For *apply*, it takes 7 days at this stage using 2,048 CPU cores with a one-hour timeout for each theorem. The substantial time cost is attributed to the $O(n^2)$ complexity of our algorithm and the memory-intensive characteristics of *Leandojo*. We believe this overhead could be greatly reduced through a more meticulous implementation. After retrieving the invocable theorems, we construct new statements and proofs for the target theorems in approximately an hour using 24 CPU cores. We then write back the mutated theorems and compile the enlarged repository through *lake build* [8], utilizing 2,048 CPU cores. We retrieve the error messages returned by Lean, which can be parsed to locate the wrong theorems. Finally, we trace the enlarged repository on a 96-core machine for 3 days, obtaining the additional state-tactic pairs by parsing the AST of each file.

**Model Training**. We select *Llama-3-8B* (Dubey et al., 2024) and *deepseek-coder-base-v1.5- 7B* (Guo et al., 2024) as our base models. We conduct continual pretraining with the next-token prediction objective for one epoch. Then we fine-tune the models with the proofstep prediction objective (Polu & Sutskever, 2020) for two epochs. All experiments are conducted on $8 \times H100$ GPUS. We employ a linear learning rate scheduler with a 3% warm-up period and a maximum learning rate of 2e-5. We set the global batch size to 256 and the cutoff length to 2,048. All models are trained using *Deepspeed ZeRO Stage3* (Rajbhandari et al., 2021) and *Flash-Attention 2* (Dao, 2024). We utilize the open-sourced codebase *Llama-Factory* (Zheng et al., 2024) for all training experiments.

**Evaluation**. We follow the evaluation setting used in Azerbayev et al. (2024). We use best-first-search as our search tragedy with a 10-minute timeout. The search budget is represented as $attempt \times sample \times step$. Here $attempt$ denotes the number of attempts, $sample$ denotes the number of generated tactics per iteration, and $step$ denotes the maximum number of steps per attempt. We choose $1 \times 32 \times 100$ as our search setting. The evaluation script is modified from an open-source implementation (Welleck, 2023) which is based on *vLLM* (Kwon et al., 2023) and *Leandojo* (Yang et al., 2023). We utilize *Leandojo Benchmark* (Yang et al., 2023) which contains 2,000 theorems as the test split of Mathlib4 and report the results on both the *random* split and the *novel_premises* split. We remove the subsets of theorems for both splits that can not be initialized by *Leandojo*. There remain 1,929 theorems in *random* split and 1,659 theorems in *novel_premises* split. We upgrade the tool-chain version of miniF2F (Zheng et al., 2022) to *v4.6.0 rc1*.

### 4.2 ANALYSIS OF SYNTHETIC DATA

We separately run the synthesis pipeline for these two tactics. For *rw*, we choose Mathlib theorems as candidate theorems. Additionally, candidate theorems for *apply* should have at least one explicit assumption. In practice, the synthesis process is divided into two stages. In the first stage, we find

---

[4]commit: 3c307701fa7e9acbdc0680d7f3b9c9fed9081740

[5]version: 1.7.1

[6]lean-toolchain: v4.6.0 rc1

[7]512 CPU nodes, each node has 8 cores and 56GB RAM

[8]https://github.com/leanprover/lean4/blob/master/src/lake/README.md

Table 2: Number of theorems. Stage one: the number of invocable instructions for all candidate theorems. Stage two: the number of theorems that pass the verification of the Lean theorem prover.

| Tactic | Candidate theorems | Stage one | Stage two | Expansion | Conversion Ratio |
|---|---|---|---|---|---|
| rw | 110,657 | 5,081,544 | 2,830,817 | ×25 | 56% |
| apply | 78,871 | 9,483,504 | 3,495,832 | ×44 | 37% |

the potential invocable theorems for each candidate theorem by running a specific tactic. In the second stage, we construct the new theorems and verify their correctness using the Lean theorem prover. Table 2 shows the number of theorems of different stages. For both tactics, we increase the number of theorems by an order of magnitude (×25 for *rw* and ×44 for *apply*). The conversion ratios from the potential invocable theorems to the outcomes are primarily determined by the method used to construct the new statements and proofs. We believe that a finer implementation could greatly improve the conversion ratio. Fig.3 shows the dynamics of the distribution of mathematical subjects. The *rw* tactic increases the percentages of Analysis, Ring Algebra, Number Theory, and so on. The *apply* tactic mainly contributes to the fields of Analysis and Topology. Further information about synthetic data can be found in the Appendix D.

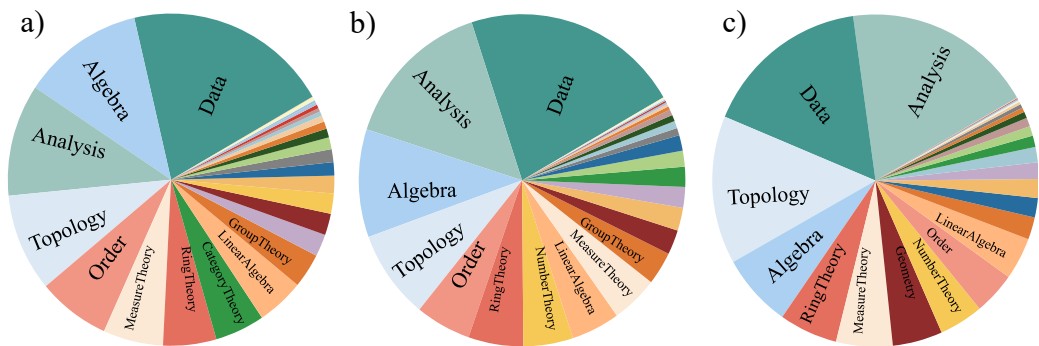

Figure 3: Distribution of mathematical subjects. For each employed tactic, we mix the generated variants with the original theorems. a) The distribution of Mathlib. b) The distribution of Mathlib + *rw*. c) The distribution of Mathlib + *apply*.

Our method synthesizes a large collection of new theorems utilizing each tactic. Then we combine them with the theorems in Mathlib as the training data for continual pre-training. Our approach also introduces new state-tactic pairs during the theorem-construction process. We write the variants to corresponding lean files and extract additional state-tactic pairs using *Leandojo*. The synthesized data are categorized primarily based on the employed tactic, specifically *rw* and *apply*. Variants and their corresponding state-tactic pairs that appear in the test split of the *Leandojo* benchmark are removed. Furthermore, the extracted state-tactic pairs are deduplicated according to the invocable theorem (i.e., premise) used in the tactic instruction. Finally, we obtain about 30k data points for each tactic. We combine them with the training set of *Leandojo* (Mathlib-train) that composes over 200k data points to form the supervised fine-tuning dataset. A detailed description of the deduplication process and training data are presented in the Appendix D.3.

## 4.3 EXPERIMENTAL RESULTS

### 4.3.1 MAIN RESULTS

We conduct continual pretraining on the augmented lean corpus. Then we fine-tune the LLMs on the mixture of Mathlib-train and additional state-tactic pairs. The training data are grouped by the tactic employed in the additional state-tactic pairs. We evaluate the effectiveness of our method on the challenging *Leandojo* benchmark and report results on different mixtures of data. As shown in Table 3, our synthetic data consistently improve the theorem-proving capabilities of LLMs. Compared with solely finetuning on the training split of Mathlib, data augmentation for a single tactic

Table 3: Results on Mathlib. tidy: a tactic in Mathlib that uses heuristics to complete a proof. The results of tidy and GPT4 were reported in Yang et al. (2023). We select the performance of each model solely fine-tuned using Mathlib-train as the main baseline. Mathlib-train + x: the performance of the model pre-trained and fine-tuned on a mixture of Mathlib-train and additional data about x.

| Methods | random | novel_premises | Search Budget |
|---|---|---|---|
| tidy | 23.8 | 5.3 | - |
| GPT-4 | 29.0 | 7.4 | $1 \times 35$ |
| Reprover (Yang et al., 2023) | 47.6 | 23.2 | $1 \times 64$ |
| w/ retrieval | 51.2 | 26.3 | $1 \times 64$ |
| llmstep (Pythia 2.8b) (Welleck & Saha, 2023) | 47.6 | - | $1 \times 32$ |
| | 50.1 | - | $2 \times 32$ |
| *Llama3-8b* | 58.22 | 38.52 | $1 \times 32$ |
| Mathlib-train + rw | 59.62 (+1.40) | 42.13 (+3.62) | $1 \times 32$ |
| Mathlib-train + apply | 58.84 (+0.62) | 41.29 (+2.77) | $1 \times 32$ |
| Mathlib-train + rw + apply | **59.82 (+1.60)** | **43.22 (+4.70)** | $1 \times 32$ |
| *deepseek-coder-7b-base-v1.5* | 57.7 | 39.24 | $1 \times 32$ |
| Mathlib-train + rw | 59.25 (+1.55) | 42.98 (+3.74) | $1 \times 32$ |
| Mathlib-train + apply | 58.68 (+0.98) | 40.51 (+1.27) | $1 \times 32$ |
| Mathlib-train + rw + apply | **60.39 (+2.69)** | **43.46 (+4.22)** | $1 \times 32$ |

demonstrates a beneficial effect on the theorem-proving ability of LLMs. Moreover, the positive impacts of each tactic can be cumulative. Training on the combination of *rw* variants and *apply* variants results in a significant performance improvement in the challenging novel_premises split of *Leandojo* benchmark, where the model is required to use at least one new premise to prove the target theorem (+4.70%, 78 theorems for *Llama3-8b*; +4.22%, 70 theorems for *deepseek-coder-7b-base-v1.5*). Our synthetic data still make a certain improvement on the random split, where the performance of models is over-estimated by allowing it to prove many theorems through memorization. In conclusion, the results of the experiment show that simply mutating the seed theorems and introducing state-tactic pairs of a single tactic can relieve the data scarcity problem and enhance the theorem-proving ability of LLMs.

### 4.3.2 EFFECTIVENESS OF CONTINUAL PRETRAINING

Table 4: Effectiveness of continual pre-training. We grouped the dataset for CPT and SFT by the tactic employed in the additional state-tactic pairs.

| Methods | random | novel_premises | random | novel_premises |
|---|---|---|---|---|
| | *Llama3-8b* | | *deepseek-coder-base-7b-v1.5* | |
| *sft: mathlib-train* | | | | |
| w/o cpt | 58.22 | 38.52 | 57.70 | 39.24 |
| rw | 59.56 (+1.34) | 42.56 (+4.04) | 58.74 (+1.04) | 40.69 (+1.45) |
| apply | 58.42 (+0.21) | 41.29 (+2.77) | 58.58 (+0.88) | 40.02 (+0.78) |
| rw + apply | 59.72 (+1.50) | 42.19 (+3.67) | 59.67 (+1.97) | 41.65 (+2.41) |
| *sft: mathlib-train + rw* | | | | |
| w/o cpt | 57.85 | 41.59 | 58.63 | 41.05 |
| rw | 59.62 (+1.77) | 42.13 (+0.54) | 59.25 (+0.62) | 42.98 (+1.93) |
| *sft: mathlib-train + apply* | | | | |
| w/o cpt | 56.71 | 40.02 | 57.96 | 41.17 |
| apply | 58.84 (+2.13) | 41.29 (+1.27) | 58.68 (+0.72) | 40.51 (-0.66) |
| *sft: mathlib-train + rw + apply* | | | | |
| w/o cpt | 58.53 | 41.95 | 58.37 | 42.92 |
| rw + apply | 59.82 (+1.29) | 43.22 (+1.27) | 60.39 (+2.02) | 43.46 (+0.54) |

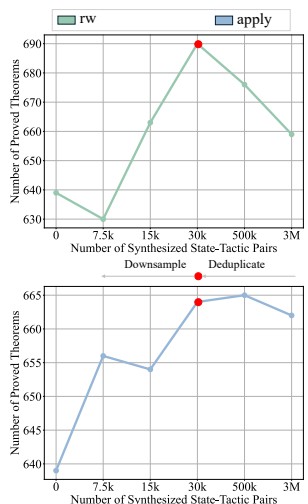

Figure 4: Influence of the quantity of synthesized data points.

To examine the necessity of continual pretraining, we assess and contrast the performance of the LLM on *Leandojo* benchmark when the pretraining stage is included versus when it is excluded from the experimental setup. We use models fine-tuned on various combinations of state-tactic pairs as our baselines and present the results of pretraining on the augmented corpus. As shown in Table 4, the continual pretraining stage demonstrates a positive influence on the performance of LLMs across diverse supervised fine-tuning settings. The experimental results indicate that continual pretraining before the supervised finetuning stage is also beneficial to the theorem-proving ability of the LLM.

### 4.3.3 INFLUENCE OF THE QUANTITY OF SFT DATASET

We deduplicate the synthesized state-tactic pairs of each tactic by the invocable theorem (i.e., premise). Then we obtain about 30k data points for each tactic. To examine the influence of the quantity of the SFT dataset, we compare the performance of *Llama-3-8B*, trained on different quantities of additional data points, on novel_premises split of *Leandojo* benchmark. As shown in Fig.4, the selected quantity (30k) achieves a relatively optimal compromise between the performance and overhead. The experimental results also reveal that enlarging the quantity of state-tactic pairs of a single tactic tends to lead to rapid saturation. We assume that the key to continually improving the theorem-proving ability lies in keeping the diversity of tactics during the process of scaling the synthetic data. More details are presented in Appendix D.3.4.

### 4.3.4 ANALYSIS OF OUT-OF-DISTRIBUTION PERFORMANCE

We evaluate *Llama-3-8b* using the competition-level theorem proving benchmark miniF2F. As shown in Table 5, our synthesized data still helps to improve the theorem-proving ability of LLMs on the out-of-distribution benchmark. The magnitude of this improvement is comparatively smaller than that observed on the in-distribution benchmark. We attribute this discrepancy to the divergence between synthesized tactics and the preferred tactics to prove competition-level problems. Through manual inspection of the correct proofs generated by various LLMs trained on Mathlib-train, we identify a tendency to favor advanced and automated tactics (e.g., *simp*, *omega*, *linarith*, *norm_num*, etc.). Additionally, we analyze the distribution of tactics used in proved theorems across different data compositions and make the following observations: 1) Data augmentation on a single tactic will increase the model's preference for the specific tactic; 2) Adjusting the distribution of different tactics within the dataset is promising to improve the theorem-proving ability of LLMs. The entire analysis process is illustrated in Appendix E.2.

Table 5: Results on miniF2F. We evaluate the performance across different data compositions and list the ratio of *rw*, *apply*, *norm_num* and *linarith* used by Llama3-8b to prove these theorems.

| Methods | miniF2F-test | Correct/Total | *rw* | *apply* | *norm_num* | *linarith* |
|---|---|---|---|---|---|---|
| Mathlib-train | 34.01 | 83/244 | 16.10 | 0.00 | 27.12 | 16.95 |
| Mathlib-train + rw | 35.24 | 86/244 | 18.75 | 0.78 | 14.84 | 21.88 |
| Mathlib-train + apply | 36.07 | 88/244 | 8.87 | 2.42 | 20.16 | 15.63 |
| Mathlib-train + rw + apply | **36.48** (+2.47) | 89/244 | 12.31 | 0.77 | 26.92 | 16.92 |

## 5 CONCLUSION

We have presented a general data-synthesis framework for the Lean theorem prover, which amplifies the theorem-proving capability of the LLM through symbolic mutation. Our algorithm increases the number of theorems in Mathlib by an order of magnitude and achieves promising results in improving the theorem-proving ability of the LLM. Synthesizing formal theorems is an inherently challenging problem. Our approach, much like ancient alchemy, involves experimenting with a substantial number of theorems in the hope of uncovering valuable "gold". We aspire for our algorithm to serve as a foundation for further research, advancing theorem synthesis from alchemy to chemistry.

ACKNOWLEDGMENTS

This research was supported by the National Natural Science Foundation of China (No. U23B2060, No.62088102). We sincerely thank the Lean Community for providing help about this work. We also appreciate the anonymous reviewers for their helpful comments.

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

CONTENTS

## A  BACKGROUND ON LEAN

Lean (de Moura et al., 2015) is a functional programming language and interactive theorem prover based on dependent type theory. As one of the most popular formal systems, Lean aids mathematicians in formalizing statements and proofs in a semi-auto style and enables them to verify the correctness of each proof step through rigorous type-checking.

**Theorem in Lean**. To some extent, theorems in Lean can be seen as a special variant of functions in general-purpose programming languages. A theorem consists of a statement and corresponding proof. In Lean, the keyword "theorem", "example" or "lemma" is used to define the "function", sometimes followed by a specific function name. The assumption of a statement can be formatted as implicit or explicit arguments, while the assertion of the statement specifies the return type of the function. The proof of the statement can be viewed as the function body, which constructs a proof term with the type specified by the assertion. There are two main proof styles in Lean: term-style and tactic-style. In term-style proofs, theorems are proven using constructive methods. On the other hand, tactic-style proofs sequentially decompose the proof goal using specific tactics. Although tactic-style proofs are less readable, they tend to have shorter proof lengths. Most machine learning-based theorem-proving systems focus on tactic-style proof. The synthesis method proposed by our paper can be applied to both styles.

**Tactic**. Lean offers various advanced tactics for theorem proving, which set it apart from other formal systems (e.g., Coq, Isabelle). In handwritten proofs, authors tend to guide the reader on building the proof through instructions such as "apply the previous lemma", "invoke the principle of mathematical induction", or "simplify the expression". Similarly, tactics in Lean are used to describe how to construct a proof term incrementally. They help users decompose the proof goal step by step, allowing users to focus on only one proof goal at a time.

**Mathlib**. Mathlib[9] is a comprehensive mathematical library for Lean, largely maintained by the community, which encompasses a broad spectrum of mathematical subjects (e.g., algebra and analysis) and consists of over 110,000 theorems along with their respective axioms and definitions. This extensive knowledge base serves as the primary corpus for neural theorem provers.

## B  LIMITATIONS

Our method exhibits some limitations that remain to be addressed in future endeavors.

**Data Diversity and Quality**. We only define two symbolic rules (using two tactics) to synthesize new theorems. The implementation of the synthesis pipeline is over general and utilizes little domain knowledge, which affects the diversity and quality of synthetic data.

**The Cost of Synthesizing**. Despite the CPU-only nature of our algorithm, the cost of synthesizing remains huge. We believe the overhead can be significantly reduced with a finer implementation and more specialized tools to interact with the Lean theorem prover.

**Single-Round v.s. Multi-Round**. Theoretically speaking, our algorithms can be iteratively executed by adding the synthesized theorems into seed theorems. Conversely, the synthesized repository is very heavy, which makes it hard to interact with Lean using *Leandojo* and deploy our algorithm on existing hardware.

**Theorem-level or Term-level**. Our method synthesizes theorems from top to bottom and introduces additional state-tactic pairs of specific tactics. Synthesizing formal data at the theorem level is not efficient and not consistent with the step-by-step nature of theorem-proving. Ideally, we anticipate that we can synthesize formal data directly at the term level, which aligns with the characteristics of interactive theorem proving.

**Up-to-down v.s. Down-to-up**. We synthesize theorems in an up-to-down fashion. We construct the new statements first and then retrieve the correct proofs. The up-to-down fashion depends on a specific set of seed theorems, which restricts the diversity of synthetic data. A more fundamental idea is that we can sample some terms in the symbolic space directly, merge them using symbolic manipulations, and then find the corresponding goals for this new theorem. This *AlphaGeometry-*

---

[9]https://github.com/leanprover-community/mathlib4

style idea is hard to implement in Lean and requires a large amount of domain knowledge and engineering endeavors.

**Symbolic Synthesis in Conjunction with Other Techniques**. Our proposed method demonstrates significant potential for integration with other techniques to enhance the theorem-proving capabilities of LLMs. We posit that theorem synthesis in the symbolic space serves as a valuable complement to prevailing auto-formalization methods. For instance, it may contribute to the expansion of autoformalized datasets. Besides, our approach generates a substantial quantity of new proven statements which can be utilized as a comprehensive database for Retrieval-Augmented Generation (RAG) (Yang et al., 2023; Wang et al., 2024). Our objective is to amalgamate these methodologies to develop a robust theorem prover in the future.

## C  DETAILED INFORMATION OF SYNTHESIZING ALGORITHMS

### C.1  OVERVIEW

As discussed in Section 3, the entire algorithm is composed of four steps: 1) Find invocable theorems for the candidate theorem by executing a specific tactic and retrieving the resulting proof state; 2) Construct new statements, where we parse the resulting proof state and mutate the old statement with the help of AST; 3) Establish the entire proof by inserting a *have* tactic and integrating it with the old proof to build the whole proof for this new statement; 4) Verify the correctness of generated theorems in Lean theorem prover. In practice, we separately run the time-consuming first step on hundreds of 8-core CPU nodes and unify step 2) and step 3) together to construct the new theorem. Then we will write back synthetic theorems and run "lake build" to verify the generated theorems.

### C.2  FIND INVOCABLE THEOREMS

For each candidate theorem, we check whether other theorems can be used to rewrite or apply to it by executing tactics. We use the *run_tac* API provided by *Leandojo* to run a specific tactic and extract the valid proof state according to predefined criteria. The instruction templates for each tactic are listed in Table1. Here is the code snippet that illustrates this process.

```
'''args:
    dojo: interactive environment
    init_state: initial proof state of target theorem
    theorem: a possible invocable theorem
    hypos: the assumptions of the target theorem (extracted by parsing
    the AST)
'''
def is_invocable_theorem(
    dojo, init_state, theorem, hypos, mode="rw"
):
    name = theorem.full_name
    if mode == "rw":
        # e.g. rw [name] at hypo_name
        insts = get_rw_insts(name, hypos)
    elif mode == "apply":
        # e.g. have hypo_str := by apply name
        insts = get_apply_insts(name, hypos)
    res = []
    for i, inst in enumerate(insts):
        try: next_state = dojo.run_tac(init_state, inst)
        except Exception as e: ...
        else:
            state_info = {
                "init_state": init_state.pp,  # pp means pretty-printed
                "next_state": next_state.error if isinstance(next_state,
    LeanError) else next_state.pp,
                "rule": inst
}
            if isinstance(next_state, LeanError):
                if mode == "implication" \
```

```
29              and "unsolved goals" in next_state.error:
30                  res.append(state_info)
31          elif isinstance(next_state, TacticState):
32                  res.append(state_info)
33      return res
```

Listing 1: Find invocable theorems by running tactics.

We set different validation criteria for each tactic. For the *rw* tactic, if the resulting state is a TacticState, we annotate this theorem as invocable. In contrast, for the *apply* tactic, the resulting state should be "unsolved goals". Additionally, we filter the resulting invocable theorems to simplify the problem of constructing new theorems. Specifically, we remove the invocable theorems whose next_state contains meta-variables (e.g.,$?a, ?m123$) for the *rw* tactic and unnamed meta-variables (e.g.,$?e12384$) for the *apply* tactic. Ultimately, we retrieve the invocable theorems for each candidate theorem. One example of invocable theorems is shown in Fig.5.

---

**theorem_name**: Char.ofNat_toNat
**rule**:   have h : isValidCharNat c.toNat := by apply List.rel_of_pairwise_cons
**init_state**:
c : Char
h : isValidCharNat (toNat c)
⊢ ofNat (toNat c) = c
**next_state**:
unsolved goals
case hp
c : Char
h : isValidCharNat (toNat c)
⊢ Std.RBNode.All isValidCharNat ?t

case H
c : Char
h : isValidCharNat (toNat c)
⊢ ∀ {x : ℕ}, x ∈ ?lb → isValidCharNat x

case a
c : Char
h : isValidCharNat (toNat c)
⊢ Std.RBNode.lowerBound? ?cut ?t ?lb = some (toNat c)

case lb
c : Char
h : isValidCharNat (toNat c)
⊢ Option ℕ

case cut
c : Char
h : isValidCharNat (toNat c)
⊢ ℕ → Ordering

case t
c : Char
h : isValidCharNat (toNat c)
⊢ Std.RBNode ℕ

---

Figure 5: Examples of invocable theorems for *apply*

The experiments run on a large collection of CPUs ($512\times8$-core for the *rw* tactic and $256\times8$-core for *apply*). The substantial CPU requirement is largely due to the memory-intensive nature of *Leandojo*, which hinders multiprocessing on a single node. We anticipate a significant reduction in the cost of

our experiments by implementing a lighter interface for Lean interaction. The operation of *apply* is more complex and time-consuming than *rw*. We set a one-hour timeout for each *dojo* environment to reduce the time cost. When running a specific tactic, we do not add additional imports to the *dojo* environment to avoid introducing human preferences in the process of synthesis. This setting may narrow the scope of theorems that the tactic can access and lower the variety of invocable theorems.

In summary, finding invocable theorems constitutes the most time-consuming and computationally intensive stage of our algorithm, entailing trade-offs among cost, time, and generated data volume.

### C.3 CONSTRUCT NEW THEOREMS

To create a new theorem, we construct the new statement using the invocable theorems returned by Section C.2 and then establish the entire proof through *have* tactic. Our symbolic engine is built upon *Leandojo* API, utilizing the extracted AST and some string manipulations. To facilitate the detailed explanation of algorithms, we will delineate the implementation of these two tactics separately in the following pseudocode or source code.

#### C.3.1 *rw* TACTIC

The logic of constructing a new statement for *rw* tactic is simple. We just identify whether a specific assumption or assertion has been rewritten by parsing invocable instructions with regular expressions. Then we parse the AST node of the candidate statement to locate the corresponding part that should be mutated. Finally, we extract the new assumption or assertion from the next proof state and replace the old one with the new one. The main procedure is shown in Algorithm 2.

---

**Algorithm 2** Construct new statement for rw tactic

---

**Input:** candidate statement $s$, invocable theorem $t_{inv}$
**Output:** mutated statement $s_m$
$node \leftarrow \text{EXTRACT\_AST}(s)$               ▷ extract the AST of candidate statement
$\_, next\_state, inst \leftarrow t_{inv}$             ▷ get the next state and instruction
flag $\leftarrow \text{IDENTIFY}(i)$    ▷ flag specifies whether the assumption or assertion should be mutated
location $l \leftarrow \text{PARSE}(node, t_{inv}, flag)$    ▷ parse AST node and locate the corresponding part that should to be mutated
mutant $m \leftarrow \text{CONSTRUCT}(next\_state)$    ▷ parse the next proof state and construct the target string
new statement $s_m \leftarrow \text{REPLACE}(s, m, l)$

---

After creating a new statement, we should insert a *have* tactic to construct the whole proof. If the assumption is modified, then we just restore it to the old one by reversing the direction of *rw* within a *have* instruction and then concatenate it with the original proof. If the assertion is mutated, the *have* tactic can be used to prove the original assertion with initial proof block. Then we just rewrite the old proof goal to the new one to construct the whole proof. Here is a simplified code snippet that illustrates this process.

```
1  def proof_generation_rw(
2          invocable_inst,
3          flag,
4          proof_str,
5          conc_or_hypo_old=None,
6          is_tactic_style=False
7          ):
8      inst = invocable_inst["rule"]
9      if flag == "hypo":
10         hypo_name = parse(inst, flag)
11     # find the delimiter for proof str(e.g. := by or :=)(simplified
       version)
12     if is_tactic_style:
13         delimiter = ":= by"
14     else:
15         delimiter = ":="
```

```
16      splits = proof_str.split(delimiter)
17      proof_seqs = delimiter.join(splits[1:])
18      if flag == "hypo":
19          rev_inst = reverse_rw(invocable_inst)
20          have_template = "have {subgoal} := by {proof_seqs}"
21          have_inst = have_template.format(
22              subgoal=conc_or_hypo_old,
23              proof_seqs=rev_inst)
24          have_inst += f';exact {hypo_name}'
25          end_inst = proof_seqs
26      elif flag == "conclusion":
27          have_template = "have : {subgoal} {delimiter} {proof_seqs}"
28          have_inst = have_template.format(
29              subgoal=conc_or_hypo_old,
30              delimiter=delimiter,
31              proof_seqs=proof_seqs)
32          head = "by " if not is_tactic_style else ""
33          _suffix = " at this;exact this"
34          end_inst = head + inst + _suffix
35      # do indentation
36      have_inst = indent_code(delimiter, proof_str, have_inst, indent_level
        =...)
37      end_inst = indent_code(delimiter, proof_str, end_inst, indent_level
        =...)
38      # concat the different parts of proof
39      prefix = splits[0] + delimiter + '\n'
40      suffix = end_inst if end_inst.startswith('\n') else '\n' + end_inst
41      new_proof = prefix + have_inst + suffix
42      return new_proof
```

Listing 2: Build the whole proof for *rw* tactic

### C.3.2 *apply* TACTIC

---

**Algorithm 3** Construct new statement for apply tactic

---

**Input:** candidate statement $s$, invocable theorem $t_{inv}$
**Output:** mutated statement $s_m$
$H \leftarrow \emptyset$          ▷ initialize the set of new assumptions
$node \leftarrow \text{EXTRACT\_AST}(s)$      ▷ extract the AST of candidate statement
$\_, next\_state, inst \leftarrow t_{inv}$      ▷ get the next state and instruction
$Metavs, Goals \leftarrow \text{PARSE}(next\_state)$    ▷ get the set of metavaribales and other subgoals
**for** $metav \in Metavs$ **do**      ▷ Assigning metavariables
    Add $\text{ASSIGN}(metav, next\_state)$ to $H$
**end for**
**for** $goal \in Goals$ **do**      ▷ Fill the other subgoals depending on meta-varibales
    Add $\text{ASSIGN}(goal, next\_state, Metavs)$ to $H$
**end for**
$H \leftarrow \text{HANDLE\_NAMING\_CONFLICTS}(H)$
new assumption $h_m \leftarrow \text{CONCAT}(H)$
location $l \leftarrow \text{PARSE}(node, t_{inv})$    ▷ parse AST node and locate the old assumption that needs to be mutated
$s_m \leftarrow \text{REPLACE}(s, h_m, l)$

---

Constructing new statements for *apply* tactic is more complex than *rw*. Applying a theorem may introduce some metavariables and new subgoals into the local context for the resulting proof state as shown in Fig.5. We assign values to the metavariables by parsing the next_state and then retrieve all subgoals containing metavariables as new assumptions. For each new assumption, we can extract its name and type from the proof state. To avoid naming conflicts, we define a set of rules to rename

the variable according to the naming conversion of Mathlib[10]. Ultimately, we concatenate all new assumptions and replace the old assumption with them. This procedure is shown in Algorithm 3.

Similarly, we can construct the entire proof for the new statement by inserting a *have* lemma. The simplified code snippet illustrates this process.

```python
def proof_generation_apply(cases_goals, inst, proof_str, is_tactic_style)
    :
    if len(cases_goals) == 1:
        lemma = inst + "; assumption"
    elif len(cases_goals) > 1:
        lemma = inst + "<;> assumption"
    else:
        raise Exception("no available case and corresponding goal")

    if is_tactic_style:
        delimiter = ":= by"
    else:
        delimiter = ":="

    splits = proof_str.split(delimiter)
    proof_seqs = delimiter.join(splits[1:])
    lemma = indent_code(delimiter, proof_str, lemma, indent_level=...)
    prefix = splits[0] + delimiter + '\n'
    suffix = proof_seqs if proof_seqs.startswith('\n') else '\n' +
    proof_seqs
    new_proof = prefix + lemma + suffix
    return new_proof
```

Listing 3: Build the whole proof for *apply* tatic

.

## C.4 VERIFY THE THEOREMS

Our method creates a set of variants for each candidate theorem in Mathlib. We write the variants back to the original file and execute *lake build* for verification. We remove the wrong lines for each file by parsing the error message returned by Lean. Then, we will rebuild the repo to ensure the effectiveness of verification. We remove the files that cause errors in the rebuilding process. Specifically, for each 8-core CPU node, we only build one ".lean" file each time to speed up this process and simplify the logic of parsing. The whole experiment runs on 2,048 CPUs ($256 \times 8$-core). The code snippets illustrate the procedure for each CPU node. After verifying the correctness of the synthesized theorem, we extract the state-tactic pairs from our augmented Mathlib repository using *Leandojo*. For *rw* or *apply*, it takes three days for a 96-core CPU machine to trace the enlarged repository. In practice, we split the modified lean files into several portions, separately write them into multiple lean repositories, and trace the repos on several 96-core CPU machines.

```python
# A single 8-core CPU node
res = []
for idx, file in enumerate(files):           # for each modified file
    '''file {
            file_name: "name of the lean file",
            text:  "the content of this file after writing synthesized
    variants into this file"
            "loc": {"theorem_name": [(start_line_nb, end_line_nb)...]}
            }'''
    tmp = {
            'loc': file['loc'],
            'file_name': file['file_name'],
            'text': file['text']
            }
    file_name = file['file_name']
```

---

[10] https://leanprover-community.github.io/contribute/naming.html

```
15      file_path = os.path.join(mathlib_package_path, file_name)
16      # extract the old content of this file
17      with open(file_path, "r") as f:
18          old_str = f.read()
19      # replace the old content with new content
20      with open(file_path, "w") as f:
21          f.write(file['text'])
22      # change the build target to current file
23      with open(LIBRARY_ROOT_FILE, 'w') as f:    # LIBRARY_ROOT_PATH:
    Mathlib.lean
24          module_name = file_name.replace('/', '.').replace('.lean', '')
25          f.write(f"import {module_name}")
26      if have_variants(file):
27          ## lake build the new mathlib project
28          wd = os.getcwd()
29          result = lake_build(mathlib_package_path) #a helper function
30          os.chdir(wd)
31          ## parse the output
32          # subprocess error
33          if result == None: tmp['valid_loc'] = ["No variants"]
34          elif result == 0:
35              tmp['valid_loc'] = tmp['loc']
36              print('successful build')
37          # timeout error
38          elif result == -1: tmp['valid_loc'] = ["No variants"]
39          else:
40              # find the error locations(line numbers)
41              pattern = fr"({file_name}):(\d+):(\d+): error:"
42              errors = re.findall(pattern, result)
43              if len(errors) == 0:  tmp['valid_loc'] = ["No variants"] #
    parse exception
44              else:
45                  # extract line numbers from errors
46                  error_line_nbs = ...
47                  # get the locations of all variants
48                  intervals = ...
49                  # drop the error ones and write back
50                  valid_locs = diff(intervals, error_line_nbs)
51                  write_back(valid_locs, file['text'])
52                  ## rebuilt the project if causes error then remove this
    file
53                  wd = os.getcwd()
54                  result = lake_build(mathlib_package_path)
55                  os.chdir(wd)
56                  if result != 0:  tmp['valid_loc'] = ["No variants"] #
    rebuild error
57                  else:   # pass the rebuilding process
58                      tmp['valid_loc'] = valid_locs
59      else:
60          tmp['valid_loc'] = ['No variants']
61      # write back the original content
62      with open(file_path, "w") as f:
63          f.write(old_str)
64      res.append(tmp)
```

Listing 4: Verify the correctness of generated theorems

## C.5 LIMITATIONS OF SYNTHESIS PIPELINE

Our synthesis pipeline is mainly based on the advanced *Leandojo* tool. We use it to interact with Lean, parse abstract syntax trees and trace state-tactic pairs. However, this tool has the following weaknesses: 1) It will generate a significant number of temporary files that consume substantial disk space when initializing a "dojo" environment. The memory-intensive nature of this tool hinders our ability to effectively implement multiprocessing; 2) Moreover, it lacks native support for tracing a

local Lean repository, so we must first upload our data to GitHub; 3) We encounter challenges when tracing a repository of a scale significantly larger than that of Mathlib, which makes it hard to do multi-round synthesis. We aspire to enhance the functionality of the *Leandojo* tool to tackle more demanding scenarios in our forthcoming endeavors.

In addition, the process of constructing statements and proofs plays an important role in data volume and diversity. Our implementation involves parsing the abstract syntax tree for localization and conducting various string manipulations, which is straightforward but struggles with sophisticated situations such as coercion, naming conflicts, and other corner cases. We are looking forward to refactoring our modification logic with the metaprogramming API of lean [11] in the future, which is more robust and easier to extend.

## D   DEEPER ANALYSIS OF SYNTHETIC DATASET

### D.1   NUMERICAL ANALYSIS

The histogram of the number of variants synthesized by each tactic is shown in Fig.6.

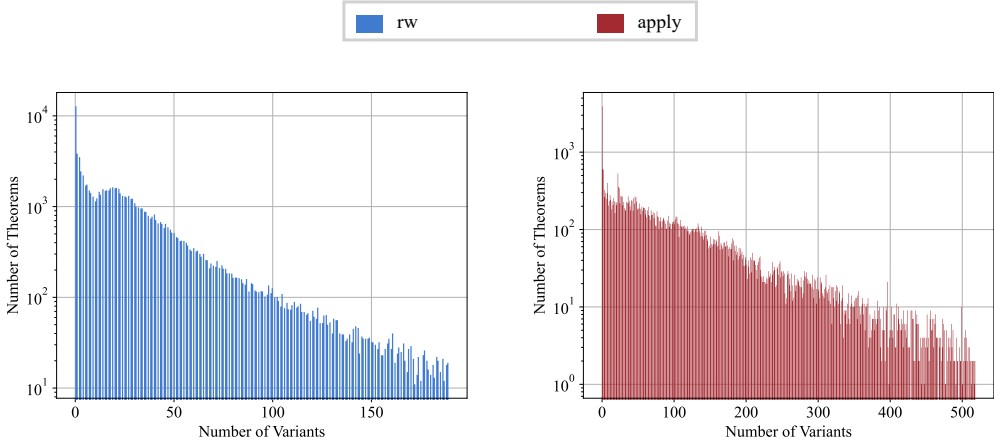

Figure 6: The distribution of the number of variants (only 99% of the data are visualized).

For each tactic, we also list the top 20 theorems with the highest number of variants in Fig.7.

### D.2   EXAMPLES

Due to the large volume of synthetic data, it is challenging to display all the data in the appendix. We only display a subset of demo theorems for reference. The proof lengths of these theorems range from 1 to 3 lines. The synthesized theorems of *rw* tactic are displayed in Fig.8. The synthesized theorems of *apply* are displayed in Fig.9.

### D.3   DETAILS OF TRAINING DATA

### D.3.1   EXAMPLES OF TRAINING DATA

As shown in Fig.10, we synthesize a series of variants for each candidate theorem by employing different tactic instructions to mutate existing theorems. We simply combine these additional theorems with the original theorems in Mathlib and train LLMs on this augmented corpus. In addition to synthesizing variants for each candidate theorem, symbolic manipulations to construct new theorems also introduce some new state-tactic pairs. What should be noted is that the state-tactic pairs are extracted by *Leandojo* rather than manually designed symbolic rules. We have not performed

---

[11]https://leanprover-community.github.io/lean4-metaprogramming-book/

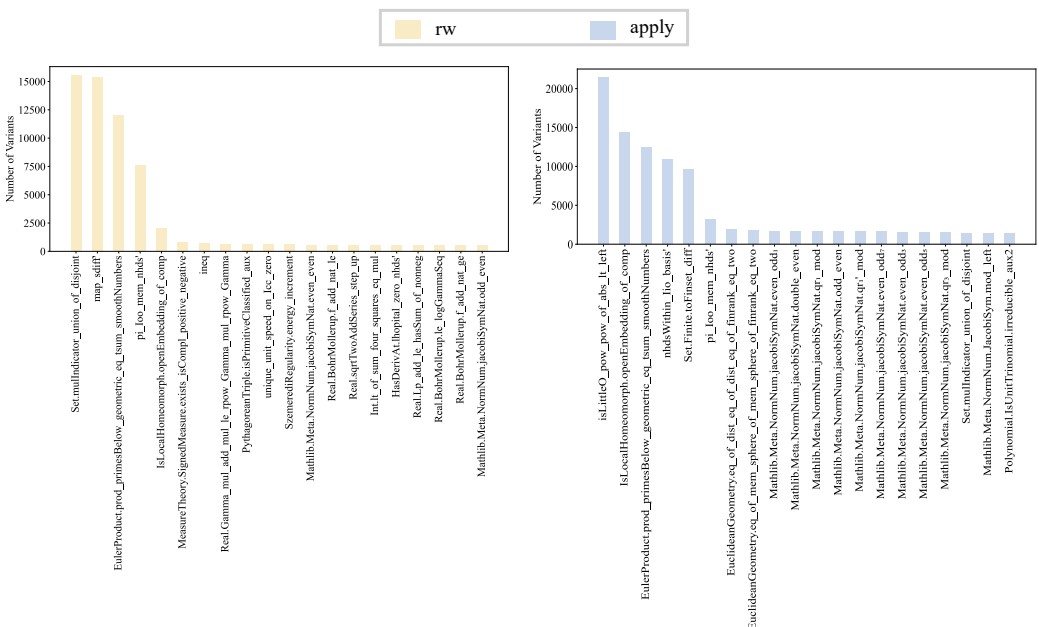

Figure 7: The top20 theorems for *rw* and *apply*.

any post-processing on the extracted state-tactic pairs. We group the extracted theorems by the employed tactics (*rw*, *apply*, *have*). The examples of *rw* and *apply* are shown in Fig.11. The examples of *have* are shown in Fig.12.

### D.3.2 PREPROCESSING

The synthesized variants of theorems and corresponding state-tactic pairs appearing in the test split of *Leandojo* benchmark are removed. During the data synthesis process, an invocable theorem may be used to rewrite or apply to different candidate theorems. Thus, many data points extracted from the augmented Mathlib repository share the same tactic and invocable theorem (i.e., premise), such as premise A in "rw [A]" or "apply A". These data points have similar changes in the proof state. We keep one state-tactic pair for each used premise in the synthesized state-tactic pairs and obtain about 30k data points for each tactic.

### D.3.3 CLASSIFICATION OF EXTRACTED TACTICS

The types of extracted state-tactic pairs are mainly determined by the symbolic manipulations to construct the theorems. We construct the proof by inserting a *have* instruction and integrating it with the original proof. As a result, we manually introduce tactics centered on *rw*, *apply* or *have*. The traced data predominantly features these tactics. The style of the seed theorem (tactic-style or term-style) and the implementation of the tracing tool are also key factors for the traced data. To see more details of this process, it is a good choice to trace the synthesized repository in person. Being familiar with the tracing process will offer some valuable guidance in designing symbolic rules to modify the proof. The extracted state-tactic pairs can also be post-processed (e.g., split the chained tactics into single ones), which has not been explored by our work.

### D.3.4 INFLUENCE OF THE QUANTITY OF SFT DATASET

We assess the impact of varying quantities of additional state-tactics pairs for each tactic under several conditions: 1) Mathlib-train with no additional data points; 2) Downsampling with a ratio of 0.25, resulting in 7.5k additional data points; 3) Downsampling with a ratio of 0.5, resulting in 15k additional data points; 4) Our setting with a deduplication threshold of 1, resulting in 30k additional data points; 5) Deduplication with a threshold of 50, resulting in 500k additional data points; and 6) No deduplication, resulting in 3M additional data points. We fine-tune Llama-3-8b on these

---

**Finset.multiplicativeEnergy_mono_right**

---

theorem multiplicativeEnergy_mono_right (ht : $t_1 \subseteq t_2$) :
  multiplicativeEnergy s $t_1$ ≤ multiplicativeEnergy s $t_2$ :=
 multiplicativeEnergy_mono Subset.rfl ht

---

example (ht : $t_1 \cap t_2 = t_1$) :
  multiplicativeEnergy s $t_1$ ≤ multiplicativeEnergy s $t_2$:=
 have ht : $t_1 \subseteq t_2$ := by rw [Finset.inter_eq_left] at ht;exact ht
 multiplicativeEnergy_mono Subset.rfl ht

example (ht : $t_1$.val $\subseteq$ $t_2$.val) :
  multiplicativeEnergy s $t_1$ ≤ multiplicativeEnergy s $t_2$:=
 have ht : $t_1 \subseteq t_2$ := by rw [←Finset.subset_def] at ht;exact ht
 multiplicativeEnergy_mono Subset.rfl ht

example (ht : $t_1 \subseteq t_2$) :
  max (multiplicativeEnergy s $t_2$) (multiplicativeEnergy s $t_1$) = multiplicativeEnergy s $t_2$:=
 have : multiplicativeEnergy s $t_1$ ≤ multiplicativeEnergy s $t_2$ :=
  multiplicativeEnergy_mono Subset.rfl ht
 by rw [←max_eq_left_iff] at this;exact this

---

**Multiset.card_le_card**

---

theorem card_le_card {s t : Multiset α} (h : s ≤ t) : card s ≤ card t :=
 leInductionOn h Sublist.length_le

---

example {s t : Multiset α} (h : s ≤ t) : ∀ {c : ℕ}, card t < c → card s < c:=
 have : card s ≤ card t :=
  leInductionOn h Sublist.length_le
 by rw [←forall_lt_iff_le'] at this;exact this

example {s t : Multiset α} (h : s ≤ t) : card s ⊓ card t = card s:=
 have : card s ≤ card t :=
  leInductionOn h Sublist.length_le
 by rw [←inf_eq_left] at this;exact this

example {s t : Multiset α} (h : s ≤ t) : card s = card t ∨ card s < card t:=
 have : card s ≤ card t :=
  leInductionOn h Sublist.length_le
 by rw [le_iff_eq_or_lt] at this;exact this

---

**Nat.one_lt_pow'**

---

theorem one_lt_pow' (n m : ℕ) : 1 < $(m + 2)$ ^ $(n + 1)$ :=
 one_lt_pow $(n + 1)$ $(m + 2)$ n.succ_ne_zero (Nat.lt_of_sub_eq_succ rfl)

---

example (n m : ℕ) : $(m + 2)$ ^ $(n + 1)$ ≠ 0 ∧ $(m + 2)$ ^ $(n + 1)$ ≠ 1:=
 have : 1 < $(m + 2)$ ^ $(n + 1)$ :=
  one_lt_pow $(n + 1)$ $(m + 2)$ n.succ_ne_zero (Nat.lt_of_sub_eq_succ rfl)
 by rw [Nat.one_lt_iff_ne_zero_and_ne_one] at this;exact this

example (n m : ℕ) : $(m + 2)$ ^ $(n + 1)$ < $(m + 2)$ ^ $(n + 1)$ * $(m + 2)$ ^ $(n + 1)$:=
 have : 1 < $(m + 2)$ ^ $(n + 1)$ :=
  one_lt_pow $(n + 1)$ $(m + 2)$ n.succ_ne_zero (Nat.lt_of_sub_eq_succ rfl)
 by rw [←Nat.lt_mul_self_iff] at this;exact this

---

Figure 8: Examples of synthesized theorems for *rw*

different mixtures of data and evaluate their performance on *random* split of *Leandojo* Benchmark. The experimental results are shown in Fig.4, demonstrating that our setting achieves a relatively optimal balance between overhead and performance.

---

**StrictMonoOn.mapsTo_Ioc**

---

lemma StrictMonoOn.mapsTo_Ioc (h : StrictMonoOn f (Icc a b)) :
   MapsTo f (Ioc a b) (Ioc (f a) (f b)) :=
 fun _c hc ↦ ⟨h (left_mem_Icc.2 <| hc.1.le.trans hc.2) (Ioc_subset_Icc_self hc) hc.1,
   h.monotoneOn (Ioc_subset_Icc_self hc) (right_mem_Icc.2 <| hc.1.le.trans hc.2) hc.2⟩

---

example (h : StrictMonoOn f (Icc a b) ↔ True) :
   MapsTo f (Ioc a b) (Ioc (f a) (f b)):=
 have h : StrictMonoOn f (Icc a b) := by apply of_iff_true; assumption
 fun _c hc ↦ ⟨h (left_mem_Icc.2 <| hc.1.le.trans hc.2) (Ioc_subset_Icc_self hc) hc.1,
   h.monotoneOn (Ioc_subset_Icc_self hc) (right_mem_Icc.2 <| hc.1.le.trans hc.2) hc.2⟩

example (H : ∀ (b_1 : Prop), (StrictMonoOn f (Icc a b) → b_1) → StrictMonoOn f (Icc a b)) :
   MapsTo f (Ioc a b) (Ioc (f a) (f b)):=
 have h : StrictMonoOn f (Icc a b) := by apply peirce'; assumption
 …
example (h : Icc a b ∈ {x | StrictMonoOn f x}) :
   MapsTo f (Ioc a b) (Ioc (f a) (f b)):=
 have h : StrictMonoOn f (Icc a b) := by apply Membership.mem.out; assumption
 …

---

**PNat.XgcdType.reduce_a**

---

theorem reduce_a {u : XgcdType} (h : u.r = 0) : u.reduce = u.finish := by
 rw [reduce]
 exact if_pos h

---

example {u : XgcdType} (h : 0 | r u) : u.reduce = u.finish:= by
 have h : u.r = 0 := by apply Nat.eq_zero_of_zero_dvd; assumption
 rw [reduce]
 exact if_pos h

example {u : XgcdType} (H : u.bp + 1 | u.ap + 1) : u.reduce = u.finish:= by
 have h : u.r = 0 := by apply Nat.mod_eq_zero_of_dvd; assumption
 …
example {u : XgcdType} (n : ℕ) (H : Nat.gcd (r u) n = 0) : u.reduce = u.finish:= by
 have h : u.r = 0 := by apply Nat.eq_zero_of_gcd_eq_zero_left<;> assumption
 …

---

**Ordnode.not_le_delta**

---

theorem not_le_delta {s} (H : 1 ≤ s) : ¬s ≤ delta * 0 :=
 not_le_of_gt H

---

example {s} (h : 0 < s) (a : 1 | s) : ¬s ≤ delta * 0:=
 have H : 1 ≤ s := by apply Nat.le_of_dvd<;> assumption
 not_le_of_gt H

example {s} (n : ℕ) (H1 : s | n) (H2 : 0 < n) : ¬s ≤ delta * 0:=
 have H : 1 ≤ s := by apply Nat.pos_of_dvd_of_pos<;> assumption
 …
example {s} (l : List ℕ) (p : List.Pairwise LE.le (1 :: l)) (a : s ∈ l) : ¬s ≤ delta * 0:=
 have H : 1 ≤ s := by apply List.rel_of_pairwise_cons<;> assumption
 …

---

Figure 9: Examples of synthesized theorems for *apply*

```
Variant of rw
theorem_name: CategoryTheory.Limits.Multicofork.sigma_condition_variant_0
file_path: Mathlib/CategoryTheory/Limits/Shapes/Multiequalizer.lean
text:
example :  MultispanIndex.fstSigmaMap I ≫ Sigma.desc (π K) ∈ [MultispanIndex.sndSigmaMap I ≫ Sigma.desc (π K)]:= by
   have : I.fstSigmaMap ≫ Sigma.desc K.π = I.sndSigmaMap ≫ Sigma.desc K.π := by
      ext
      simp
   rw [←List.mem_singleton] at this;exact this
meta: https://github.com/leanprover-community/mathlib4/commit/3c307701fa7e9acbdc0680d7f3b9c9fed9081740'

Variant of apply
theorem_name: UniformInducing.equicontinuous_iff_variant_26
file_path: Mathlib/Topology/UniformSpace/Equicontinuity.lean
text:
example {F : ι → X → α} {u : α → β} (B : Set (Set (α → β))) (s : Set (α → β)) (hB : TopologicalSpace.IsTopologicalBasis B)
(hs : IsOpen s) (h : ∀ U ∈ B, U ⊆ s → U ⊆ UniformInducing) (a : u ∈ s) :
   Equicontinuous F ↔ Equicontinuous ((u ∘ ·) ∘ F):= by
   have hu : UniformInducing u := by apply TopologicalSpace.IsTopologicalBasis.subset_of_forall_subset<;> assumption
   congrm ∀ x, ?_
   rw [hu.equicontinuousAt_iff]
meta: https://github.com/leanprover-community/mathlib4/commit/3c307701fa7e9acbdc0680d7f3b9c9fed9081740
```

Figure 10: Examples of data for pretraining

# E    ADDITIONAL EXPERIMENTS

## E.1    EFFECTIVENESS OF DIFFERENT TACTICS

We evaluate the effectiveness of different tactics by combining additional state-tactic pairs of a specific tactic with Mathlib-train and fine-tuning the LLMs using this mixture. The experimental results are shown in Table 6. We observe that state-tactic pairs of *rw* and *apply* are beneficial for the theorem-proving ability of the LLM. And the highest improvement is achieved by the combination of these two tactics. For the state-tactic pairs of *have*, we assume that these data will teach the model to introduce lemmas in the process of proving a theorem, helping them to prove the theorems in multiple steps. However, experimental data show that *have* has complex effects on the proving capacity of LLMs. The performance on a mixture of "have" and other tactics shows poorer results compared to that on a single tactic. We hope to investigate the effectiveness of *have* tactic soon.

## E.2    ANALYSIS OF THE TACTICS TO PROVE MINIF2F THEOREMS

### E.2.1    PREFERENCE IN USED TACTICS

To see the preference for the tactics used to prove competition-level problems, we perform a comprehensive analysis of the theorems proved by different LLMs. Specifically, we fine-tune different LLMs with the random train-split of *Leandojo* benchmark and gather all theorems proved by these models. The collection of these models proves 100 theorems out of 244 theorems (41%) on the test split of *miniF2F* benchmark. The average length of the proofs generated by these models is 1.38. And the distribution of these proved theorems is shown in Fig.14. We have the following observations: 1) About half of the theorems in the miniF2F test split can be proven with only 1-2 line proofs; 2) Most of the theorems are proved with advanced and automatic tactics in Lean (e.g., *norm_num*, *linarith*, *omega*, *simp*, etc.). We assume that these tactics play an important role in the theorem-proving ability of LLMs to prove competition-level problems. From the above observations, we assume that synthesizing advanced tactic data points rather than basic data points featuring *rw* and *apply* is promising to improve the performance of proving competition-level problems.

### E.2.2    INFLUENCE OF ADDITIONAL TACTICS

We analyze the distribution of used tactics in proven miniF2F problems across different data compositions. The dynamics of distribution changes are shown in Fig.15. We assume that increasing the

Table 6: The effectiveness of different tactics

| Methods | random | novel_premises | Search Budget |
|---|---|---|---|
| ***Llama3-8b*** | | | |
| Mathlib-train | 58.22 | 38.52 | $1 \times 32$ |
| ***rw tactic*** | | | |
| Mathlib-train + rw | 57.85 (-0.37) | 41.59 (+3.07) | $1 \times 32$ |
| Mathlib-train + have | 58.27 (+0.05) | 41.29 (+2.77) | $1 \times 32$ |
| Mathlib-train + rw + have | 57.96 (-0.26) | 41.53 (+3.01) | $1 \times 32$ |
| ***apply tactic*** | | | |
| Mathlib-train + apply | 56.71 (-1.51) | 40.02 (+1.51) | $1 \times 32$ |
| Mathlib-train + have | 57.44 (-0.78) | 39.24 (+0.72) | $1 \times 32$ |
| Mathlib-train + apply + have | 57.23 (-0.99) | 38.34 (-0.18) | $1 \times 32$ |
| ***both tactic*** | | | |
| mathlib-train + rw + apply | 58.53 (+0.31) | 41.95 (+3.44) | $1 \times 32$ |
| ***deepseek-coder-7b-base-v1.5*** | | | |
| Mathlib-train | 57.7 | 39.24 | $1 \times 32$ |
| ***rw tactic*** | | | |
| Mathlib-train + rw | 58.63 (+0.93) | 41.05 (+1.81) | $1 \times 32$ |
| Mathlib-train + have | 58.11 (+0.41) | 39.06 (-0.18) | $1 \times 32$ |
| Mathlib-train + rw + have | 58.74 (+1.04) | 40.57 (+1.33) | $1 \times 32$ |
| ***apply tactic*** | | | |
| Mathlib-train + apply | 57.96 (+0.26) | 41.17 (+1.93) | $1 \times 32$ |
| Mathlib-train + have | 57.02 (-0.68) | 39.66 (+0.42) | $1 \times 32$ |
| Mathlib-train + apply + have | 58.16 (+0.46) | 39.78 (+0.54) | $1 \times 32$ |
| ***both tactic*** | | | |
| Mathlib-train + rw + apply | 58.37 (+0.67) | 42.92 (+3.68) | $1 \times 32$ |

Table 7: The results of miniF2F for different LLMs. We fine-tune each model with the random train-split of *Leandojo* benchmark and evaluate their performance on miniF2F benchmark.

| Methods | miniF2F-test |
|---|---|
| Llama-3-8b | 34.01 |
| deepseek-coder-base-7b-v1.5 | 37.70 |
| deepseek-math-7b-base | 34.42 |
| llemma-7b | 32.38 |
| mistral-7b | 32.38 |
| internlm2-math-7b | 36.06 |
| Combination | **40.98** |

diversity of synthesized tactics and adjusting the tactic distribution will be beneficial to enhance the theorem-proving ability of LLMs.

**instruction**:
/- You are proving a theorem in Lean 4.
You are given the following information:
- The current proof state, inside [STATE]...[/STATE]

Your task is to generate the next tactic in the proof.
Put the next tactic inside [TAC]...[/TAC]
-/
*rw  tactic*
**input**:
[STATE]
α : Type u_1
β : Type u_2
R : Type u_3
s t : Finset α
f : α → β
n : ℕ
inst† : DecidableEq α
this : s.card = 2 ↔ ∃ x y, x ≠ y ∧ s = {x, y}
⊢ s.card = 2 ↔ ∃ a ∈ Set.range fun x y => x ≠ y ∧ s = {x, y}, Exists a
[/STATE]
[TAC]

**output**:
*rw [←Set.exists_range_iff] at this[/TAC]*
*apply  tactic*
**input**:
[STATE]
n : ℕ
inst† : NeZero n
a b : ZMod n
w : val b ∈ HAdd.hAdd (val a) ⁻¹' Nat.le n
⊢ n ≤ val a + val b
[/STATE]
[TAC]

**output**:
*apply Set.mapsTo_preimage[/TAC]*

Figure 11: Examples of *rw* and *apply* data points for finetuning

**instruction**:
/- You are proving a theorem in Lean 4.
You are given the following information:
- The current proof state, inside [STATE]...[/STATE]

Your task is to generate the next tactic in the proof.
Put the next tactic inside [TAC]...[/TAC]
-/
*have tactic*-**rw-related**
*Case one*
**input**:
[STATE]
m n a b : $\mathbb{N}$
cop : Coprime m n
ha : a $\neq$ 0
hb : b $\neq$ Int.toNat 0
⊢ a * m + b * n $\neq$ m * n
[/STATE]
[TAC]

**output**:
*have hb : b $\neq$ 0 := by rw [Int.toNat_zero] at hb;exact hb[/TAC]*
*Case two*
**input**:
[STATE]
α : Type u_1
inst† : DecidableEq α
$\mathcal{A}$ : Finset (Finset α)
u v a† : Finset α
r : $\mathbb{N}$
huv : Equiv.swap u.card v.card = Equiv.refl $\mathbb{N}$
a : Finset α
⊢ (compress u v a).card = a.card
[/STATE]
[TAC]

**output**:
*have huv : u.card = v.card := by rw [Equiv.swap_eq_refl_iff] at huv;exact huv[/TAC]*

*have tactic-apply*-**related**
**Input:**
[STATE]
α : Type u
β : Type v
inst†¹ : DecidableEq α
inst† : Fintype α
σ : Perm α
n : $\mathbb{N}$
h : orderOf σ $\in$ {x | Nat.Coprime n x}
⊢ support (σ ^ n) = support σ
[/STATE]
[TAC]

**output**:
*have h : Nat.Coprime n (orderOf σ) := by apply Membership.mem.out; assumption[/TAC]*

Figure 12: Examples of *have* data points for finetuning

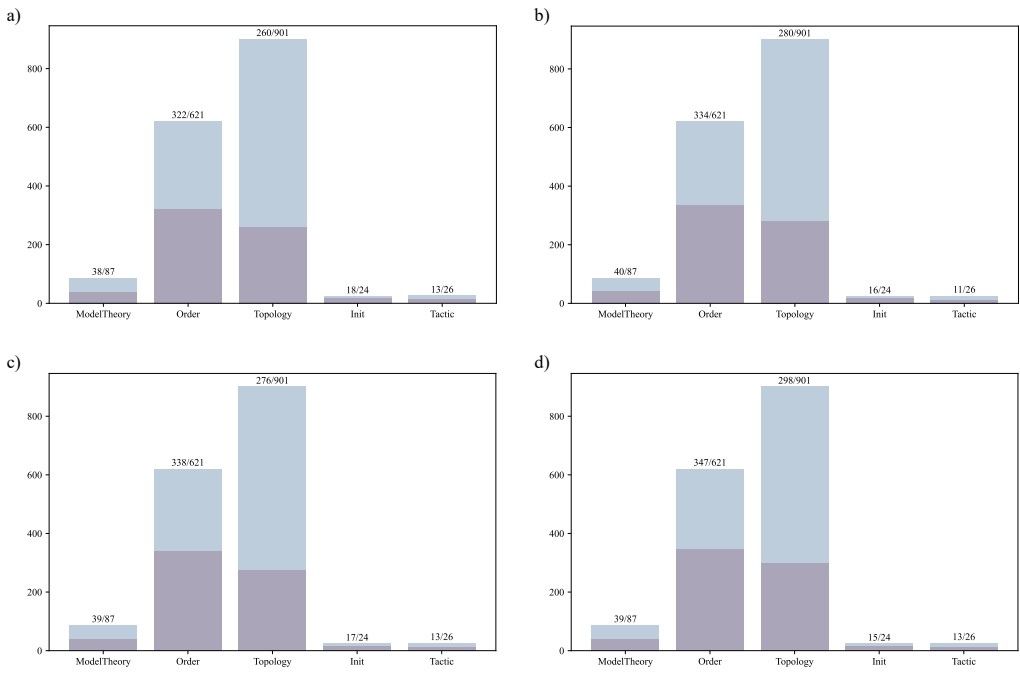

Figure 13: The performance of models fine-tuned on different SFT datasets on novel_premises split. a) Mathlib-train; b) Mathlib-train + *rw*; c) Mathlib-train + *apply*; d) Mathlib-train + rw + apply.

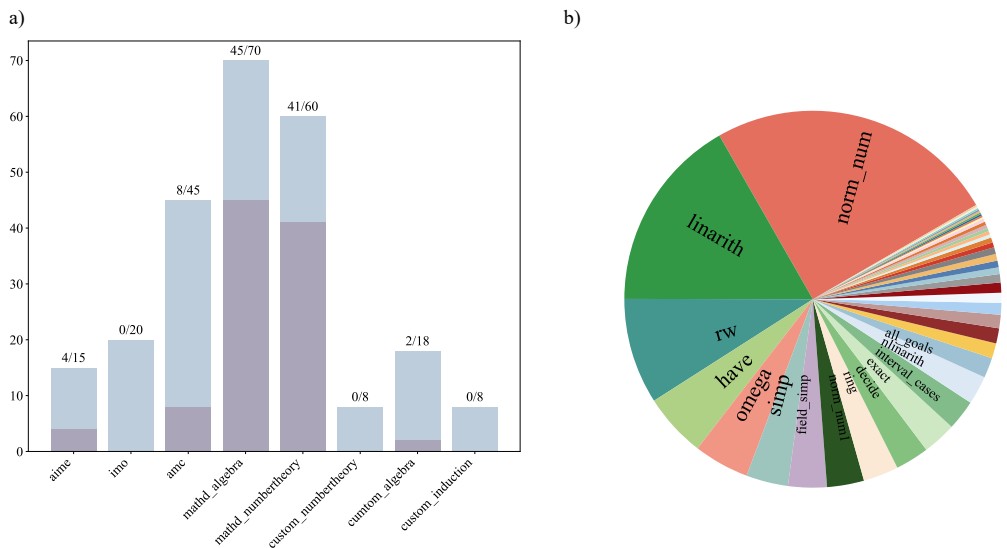

Figure 14: a) The distribution of theorems proved by different LLMs; b) The distribution of tactics used in the proved theorems.

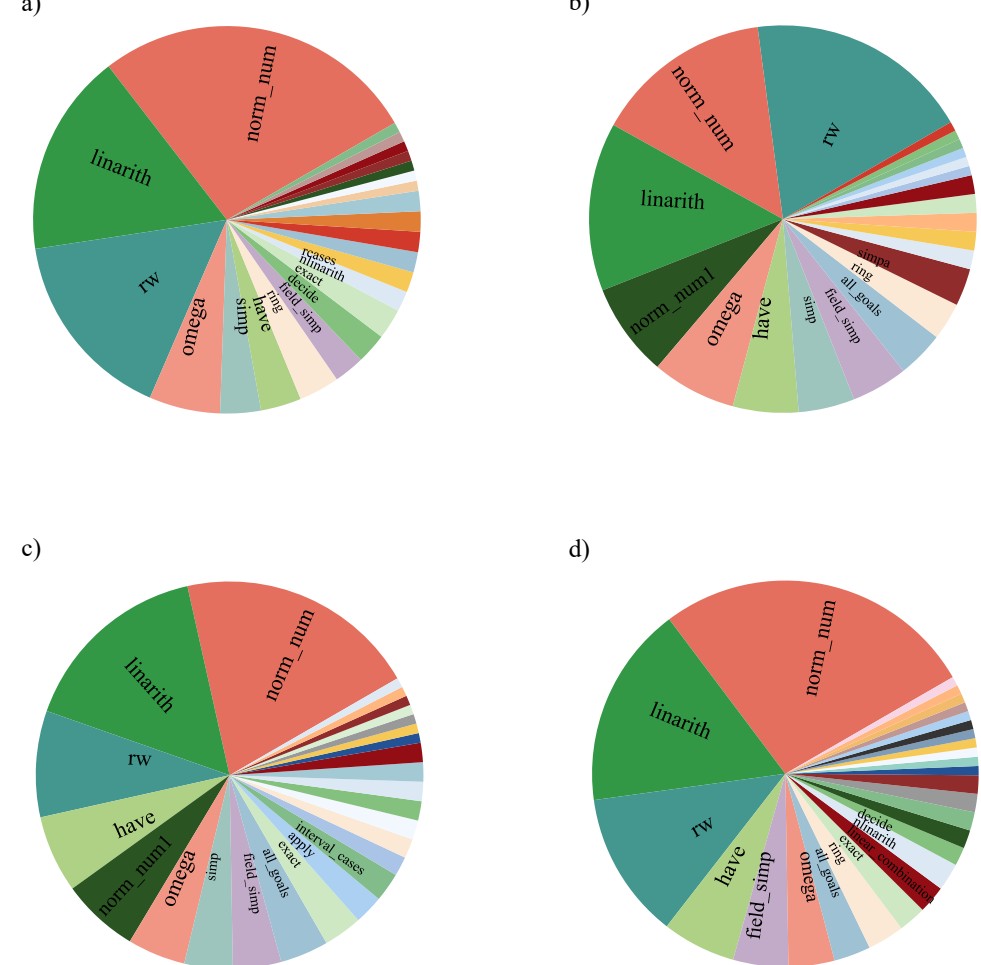

Figure 15: The distribution of used tactics for Llama-3-8b fine-tuned on different SFT datasets to prove miniF2F. a) Mathlib-train; b) Mathlib-train + *rw*; c) Mathlib-train + *apply*; d) Mathlib-train + *rw* + *apply*.

