# OpenReview forum: "Alchemy: Amplifying Theorem-Proving Capability Through Symbolic Mutation"
_ICLR.cc/2025/Conference — ICLR 2025 Poster_

### Official Review · Reviewer_NSyk · 2024-11-02

**Soundness:** 3
**Presentation:** 3
**Contribution:** 3
**Rating:** 8
**Confidence:** 3

**Summary:**

The paper proposes a new method to synthesize theorem training data for improving LLM's ability in theorem-proving. Given an existing theorem, the proposed method finds theorems that can imply its assumptions and assertions. Then, it replaces the corresponding assumptions/assertions and invokes these theorems to obtain the expanded new theorem. Experiments show the proposed method can generate 5M data and improve 7b models by a 2-4% pass rate.

**Strengths:**

- The paper is well-written and easy to understand. It has a clear motivation and proposes a novel method to generate a lot of new theorem data.

- The experimental results validate the effectiveness of the generated data. It can improve current LLMs by >4% pass rate on the novel_premises split.

**Weaknesses:**

The method seems unable to generate diverse theorem data. It mainly expands existing theorem by combining other theorems. The diversity problem may result in a lower improvement on the harder benchmark miniF2F. I guess the generated theorem can be very different from the original theorem if it has a deep variant proof tree. Authors may show the depth statistics of the generated theorem or other statistics to verify the diversity of the generated theorem.

**Questions:**

Why many generated theorems not pass the Lean prover? Since the generation process is based on symbolic replacement, I suppose most of the theorem should pass the prover.

---

> ### Author Response · Authors · 2024-11-20
> **Response to Reviewer NSyk**
>
> We want to express our sincere gratitude for your time and effort in evaluating our work. We have carefully considered your concerns and questions and attempted to give our answers.
>
> > The method seems unable to generate diverse theorem data. It mainly expands existing theorem by combining other theorems. The diversity problem may result in a lower improvement on the harder benchmark miniF2F. I guess the generated theorem can be very different from the original theorem if it has a deep variant proof tree. Authors may show the depth statistics of the generated theorem or other statistics to verify the diversity of the generated theorem.
> >
>
> We have discussed the data diversity issue and provided a metric to verify the diversity of our data in the general response. We concur with the idea of generating a broader range of theorems through multi-round mutation (deep proof tree construction), a process that may encounter numerous non-trivial challenges.
>
> > Why many generated theorems not pass the Lean prover? Since the generation process is based on symbolic replacement, I suppose most of the theorem should pass the prover.
> >
>
> We have addressed the reasons for the non-100% conversion ratio in the general response. If you have any questions, please let us know.

---

> > ### Comment · Reviewer_NSyk · 2024-11-26
> >
> > Thanks for the response. It addresses my concerns about the data diversity. I will keep my scores.

---

> > > ### Author Response · Authors · 2024-11-26
> > > **Response to Reviewer NSyk**
> > >
> > > We want to express our gratitude for your time and efforts in evaluating our work again. We deeply appreciate your acknowledgment for the value of our research.

---

### Official Review · Reviewer_BkaD · 2024-11-02

**Soundness:** 3
**Presentation:** 3
**Contribution:** 3
**Rating:** 6
**Confidence:** 4

**Summary:**

This paper introduces Alchemy, a framework to generate synthetic theorem data by applying symbolic mutations to existing theorems within Lean’s Mathlib. By mutating known theorems through symbolic operations, Alchemy expands the theorem corpus by an order of magnitude (from 110k to over 6M theorems). The authors evaluate Alchemy’s effectiveness on theorem-proving tasks, reporting a 5% improvement on the Leandojo benchmark and a 2.5% gain on the out-of-distribution miniF2F benchmark (to 36.48% test accuracy).

**Strengths:**

1. The approach is technically robust, with well-documented use of symbolic mutations (specifically rw and apply tactics) to ensure the correctness of new theorems by construction. The improvements seen on Leandojo and miniF2F benchmarks support the framework’s validity.
2. Alchemy significantly increases the number of available theorems in Mathlib, scaling up to 6 million theorems through systematic symbolic mutations. This large corpus helps address the issue of limited formal proof data for theorem-proving models. By providing a synthetic theorem corpus directly in the symbolic space, Alchemy addresses a key limitation in neural theorem proving, especially in formal languages like Lean where data is scarce and difficult to formalize manually.
3. The limitations of the approach, such as data diversity and computational cost, are clearly addressed.

**Weaknesses:**

1. Marginal Gains in Benchmark Performance: Despite generating millions of new theorems, the gains in miniF2F accuracy are limited to 2.5%, notably lower than the >60% accuracy achieved by SOTA models such as DeepSeekProver and InternLM Prover. This modest improvement raises questions regarding the utility and quality of the synthetic theorems for real-world theorem-proving tasks.
2. Computational Cost: The process of generating and verifying theorems is highly resource-intensive. The implementation reports substantial computational overhead, with 14 days on 4,096 CPU cores for rw mutations and 7 days on 2,048 cores for apply mutations, potentially limiting the accessibility and scalability of Alchemy in practice.
3. Lack of Quality Metrics for Synthetic Theorems: Although Alchemy generates a large corpus, there is limited analysis of the quality or mathematical significance of the produced theorems. Without metrics or evaluation methods beyond correctness by construction, it is challenging to assess whether the synthetic theorems provide meaningful, diverse training examples.
4. Limited Innovation Beyond Mutation: The paper relies heavily on mutating existing theorems via basic rw and apply tactics, which may restrict the variety of new insights or concepts that the synthetic data introduces. Advanced tactics (e.g., simp, linarith) and some premise selection approaches are critical in solving more challenging problems, especially in competition-level mathematics. Without these, the generated dataset might lack the depth needed to fully improve theorem-proving performance on complex out-of-distribution tasks.

**Questions:**

1. Given the modest improvement in miniF2F accuracy, are there metrics or quality checks available to assess the mathematical value or diversity of the generated theorems beyond correctness?
2. Which specific theorems in miniF2F were newly proved by the models fine-tuned with Alchemy data? This would provide insights into the areas where synthetic training data are particularly beneficial.
3. Given the computational demands, are there potential optimizations in the synthesis process to reduce the time and resources required for theorem mutation?
4. How do you avoid the data contamination problem in the evaluation/generation phase?

[1] Xin, Huajian, et al. "DeepSeek-Prover-V1.5: Harnessing Proof Assistant Feedback for Reinforcement Learning and Monte-Carlo Tree Search." arXiv preprint arXiv:2408.08152 (2024).
[2] Ying, Huaiyuan, et al. "Lean Workbook: A large-scale Lean problem set formalized from natural language math problems." arXiv preprint arXiv:2406.03847 (2024).

---

> ### Author Response · Authors · 2024-11-20
> **Response to Reviewer BkaD**
>
> We want to express our sincere gratitude for your time and effort in evaluating our work. We have carefully understood your concerns and questions. Some of them are shared concerns between reviewers. So, we answer them in the general response respectively:
>
> - **Marginal Gains in Benchmark Performance**: Poor Improvement Section of general response.
> - **Computational Cost and question-3**: High-Synthesis Cost Section of general response
> - **Lack of Quality Metrics for Synthetic Theorems and question-1:** Data Diversity Issue Section of general response
> - **question-4:** Data Contamination Section of general response
>
> We really hope our response could resolve your concerns and questions.
>
> For the individual concerns:
>
> > The paper relies heavily on mutating existing theorems via basic rw and apply tactics, which may restrict the variety of new insights or concepts that the synthetic data introduces.
> >
>
> We have discussed this possible limitation in diversity in **Appendix B** of original paper. We also provide a metric to verify the diversity of synthesized theorems in the general response.
>
> > Advanced tactics (e.g., simp, linarith) and some premise selection approaches are critical in solving more challenging problems, especially in competition-level mathematics. Without these, the generated dataset might lack the depth needed to fully improve theorem-proving performance on complex out-of-distribution tasks.
> >
>
> We have discussed the importance of advanced tactics for competition-level theorem-proving in **Section 4.3.4** and **Appendix E.2.1**. We also discuss the possibility to combine our methods with RAG in **Appendix B** to further enhance the effectiveness of our method.
>
> > Which specific theorems in miniF2F were newly proved by the models fine-tuned with Alchemy data? This would provide insights into the areas where synthetic training data are particularly beneficial.
> >
>
> We analyze the subjects of newly proved theorems by the models after fine-tuning with Alchemy data. As in the Table below,
>
> | Methods | aime | imo | amc | m-alg | m-nt| c-nt | c-alg | c-ind |
> | ---- | ---- | ---- | --- | --- | --- | --- | --- | --- |
> | Mathlib-train (original) | 2 | 0 | 5 | 40 | 35 | 0 | 1 | 0 |
> | Mathlib-train + rw  | 1 (-1) | 0 | 6 (+1) | 44 (+4) | 34 (-1) | 0 | 1 | 0 |
> | Mathlib-train + apply | 3 (+1) | 0 | 6 (+1) | 41 (+1) | 35  | 0 | 2 (+1) | 1 (+1) |
> | Mathlib-train + rw + apply | 3 (+1) | 0 | 7 (+2) | 43 (+3) | 34 (-1) | 0 | 2 (+1) | 0 |
>
> Algebra, number theory, and induction are represented by the abbreviations "alg," "nt," and "ind," respectively. Test theorems sourced from MATH and custom curation are distinguished by the labels "m" or "c."
>
> Comparing the discrepancy of distribution of solved problems for different data-compositions, we assume that rw state-tactic pairs play an important role in proving algebra problems and apply data can help for proving challenging theorems (e.g., aime, amc or custom theorems in miniF2F).

---

> > ### Comment · Reviewer_BkaD · 2024-11-24
> >
> > Nice. This comment addresses my concerns about the diversity of generated theorems and their performance. I raised my rating from 5 to 6.

---

> > > ### Author Response · Authors · 2024-11-27
> > > **Response to Reviewer BkaD**
> > >
> > > We want to express our gratitude for your kind advice during the review process. We appreciate your acknowledgement for our paper and look forward to refining our approach in the future.

---

### Official Review · Reviewer_FuWS · 2024-11-04

**Soundness:** 3
**Presentation:** 3
**Contribution:** 3
**Rating:** 6
**Confidence:** 4

**Summary:**

The paper concerns data augmentation for neural theorem proving. The authors propose a method for augmenting theorem statements and the set of (state, tactic) examples given a collection of Lean statements and proofs. Their method augments theorem statements by (1) rewriting expressions in hypotheses or the statement's goal using a rewrite tactic with a suitable premise, (2) replacing a hypothesis with a different set obtained with an apply tactic with a suitable premise. It augments proofs by undoing the rewrite and/or apply and introducing a have statement, which sometimes introduces new (state, tactic) examples.

The authors apply their augmentations to Mathlib, and finetune models with (1) continued pretraining on mathlib plus the augmented statements and proofs, followed by (2) finetuning on (state, tactic) examples from Mathlib plus those from their augmentations.

The models that have undergone continued pretraining and (state, tactic) finetuning outperform the same models when they have only undergone (state, tactic) finetuning on Mathlib alone. For example, there is a 2.69% improvement on the random LeanDojo test split, and a 4.22% improvement on the novel_premises split with DeepSeek Coder.

**Strengths:**

Originality
- The idea of synthesizing new theorem statements through rewrites and applies is new (as far as I'm aware).

Quality
- Aside from the concerns discussed below, the experiments were carried out well for several variants while adhering to standard benchmarks and search algorithm protocols.
- Implementing modifications to data extraction in Lean is likely nontrivial.

Clarity
- The data synthesis methodology was explained clearly.

Significance
- Lack of data is widely regarded as a core issue in neural theorem proving. Augmenting data using symbolic techniques is a potential approach to alleviating this issue, and the authors demonstrate a first step in this direction.
- The general direction of augmenting data using symbolic techniques is interesting and under-explored.

**Weaknesses:**

As mentioned in the strengths above, the general direction of augmenting data using symbolic techniques is interesting and under-explored. I have two primary concerns: (1) the experimental evaluation of the proposed techniques; (2) the data augmentation techniques explored in the current paper.

### Experimental evaluation
1. **Baselines**: the baseline method is a LM finetuned on (state, tactic) pairs from Mathlib. However, the proposed method does (i) continued pretraining and (ii) (state, tactic) finetuning. As a result it is difficult to interpret the main results, since there are two finetuning methodologies used. How does the baseline method perform after continued pretraining on Mathlib (without augmentation), followed by (state, tactic) finetuning on Mathlib (without augmentation)?

2. **Possible train-test overlap**: The LeanDojo benchmark consists of theorems from Mathlib. Therefore, there is potential train-test overlap in at least two places.
    - (i) First, the continued pretraining dataset, if it includes theorems from the LeanDojo test set (or premises used in the novel_premises split). How was train-test overlap prevented for continued pretraining? I wasn't able to find details on exactly what was done for continued pretraining, so it would be great to clarify this.
    - (ii) Second, the rewrites and applies may use premises that are "novel" in the novel_premises split. How do you ensure that these are not used in the data augmentation process?

As a result of (i) and (ii), it is difficult to interpret the improvement on the novel premises split. Namely, (i) and (ii) may have exposed the model to the premises required in this split, which would negate the purpose of the split. Moreover, (i) may lead to improvements on the random split as well.

3. **Finetuning hyperparameters**. This is perhaps less important than (1) and (2), but the augmented dataset leads to more gradient updates compared to finetuning on the non-augmented dataset, since finetuning is performed for a fixed number of epochs. Do the results change if the baseline is finetuned for the same number of steps as the model finetuned on the augmented dataset?

### Data augmentation techniques
1. The computational cost is very high; it takes 14 days for the rw operation on 512 CPU nodes. To make the authors' method more practical, it would have been nice to see some innovation that makes the extraction faster (either at the algorithmic level or the implementation level).

2. Currently the methods only modify the statement goal using 1 step of rewriting. The overall scientific contribution could be made stronger with more exploration of techniques (e.g., at least > 1 step of rewriting). Could you clarify why only the 1-step rewriting and apply were explored? I realize that it is hard to say how many techniques are needed (and it's always nicer to have more), so this is less of a concern for me than the experimental evaluation of the two techniques described above.

3. From what I understand, proofs are only modified by introducing a have statement that reverses the 1-step augmentation, and then the proof is the same as the original. Again it would be nice to see additional innovation in this direction.

4. It was unclear why each technique helped on unseen_premises split; could you give an intuition or an analysis of why it might help?

**Questions:**

Please see the questions above discussed in the Weaknesses. In particular, if the authors can provide a strong response to the questions regarding the experimental setup I would be willing to raise my score.

---

> ### Author Response · Authors · 2024-11-20
> **Response to Reviewer FuWS (1/2)**
>
> We are sincerely grateful to you for your comprehensive assessment. We have carefully thought about your questions and made attempts to provide answers.
>
> > **Baselines**: the baseline method is a LM finetuned on (state, tactic) pairs from Mathlib. However, the proposed method does (i) continued pretraining and (ii) (state, tactic) finetuning. As a result it is difficult to interpret the main results, since there are two finetuning methodologies used. How does the baseline method perform after continued pretraining on Mathlib (without augmentation), followed by (state, tactic) finetuning on Mathlib (without augmentation)?
> >
>
> We conjecture the Mathlib corpus is included in the baseline models’ pretraining corpus, so we didn’t continual pretrain them in our paper. According to your advice, we retrain the baseline using two finetuning methodologies. Specifically, we conduct the continual pretraining on the Mathlib (theorems in the trainset of Leandojo) and then finetuning on Mathlib-train.  The experimental results are listed in table below:
>
> | Model | random | novel_premises |
> | --- | --- | --- |
> | Llama3-8b -original | 58.22 | 38.52 |
> | Llama-3-8b-new | 57.8 (-0.42) | 39.54 (+1.02) |
> | Deepseek-Coder-7B-v1.5-original | 57.7 | 39.24 |
> | Deepseek-Coder-7B-v1.5-new | 57.91 (+0.21) | 39.54 (+0.32) |
>
> The minor improvement brought by CPT on Mathlib (without augmentation) may be attributed to Mathlib's inclusion in the pretraining data of LLMs [1, 2]. The improvements achieved on the novel_premises split is still promising (3.7% for Llama-3-8b; 3.9% for deepseek-prover-7B-v1.5).
>
> > **Finetuning hyperparameters**. This is perhaps less important than (1) and (2), but the augmented dataset leads to more gradient updates compared to finetuning on the non-augmented dataset, since finetuning is performed for a fixed number of epochs. Do the results change if the baseline is finetuned for the same number of steps as the model finetuned on the augmented dataset?
> >
>
> We conduct additional experiments on the finetuning hyperparameters. We retrain the Llama-3-8b with the same number of steps as the model finetuned on the augmented dataset. The experimental results are listed in table below:
>
> | Setting (After Mathlib CPT) | random | novel_premises |
> | --- | --- | --- |
> | original (1800 steps) | 57.8 | 39.54 |
> | current (2200 steps as in the mathlib-train + rw + apply) | 55.94 (-1.9) | 38.94 (-0.6) |
>
> The finetuning process with equal steps has not yielded the anticipated improvements for the baseline model. This outcome could be linked to unbalanced learning, as the additional 400 steps do not align with the number of steps in a single epoch.
> >Possible train-test overlap: The LeanDojo benchmark consists of theorems from Mathlib. Therefore, there is potential train-test overlap in at least two places.
> (i) First, the continued pretraining dataset, if it includes theorems from the LeanDojo test set (or premises used in the novel_premises split). How was train-test overlap prevented for continued pretraining? I wasn't able to find details on exactly what was done for continued pretraining, so it would be great to clarify this.
> (ii) Second, the rewrites and applies may use premises that are "novel" in the novel_premises split. How do you ensure that these are not used in the data augmentation process?
> As a result of (i) and (ii), it is difficult to interpret the improvement on the novel premises split. Namely, (i) and (ii) may have exposed the model to the premises required in this split, which would negate the purpose of the split. Moreover, (i) may lead to improvements on the random split as well.
> >
>
> We have provided more details about this question in the general response.
> > The computational cost is very high; it takes 14 days for the rw operation on 512 CPU nodes. To make the authors' method more practical, it would have been nice to see some innovation that makes the extraction faster (either at the algorithmic level or the implementation level).
> >
>
> We have discussed the reasons for the high cost and possible optimization methods in the general response.

---

> > ### Author Response · Authors · 2024-11-20
> > **Response to Reviewer FuWS (2/2)**
> >
> > > Currently the methods only modify the statement goal using 1 step of rewriting. The overall scientific contribution could be made stronger with more exploration of techniques (e.g., at least > 1 step of rewriting). Could you clarify why only the 1-step rewriting and apply were explored? I realize that it is hard to say how many techniques are needed (and it's always nicer to have more), so this is less of a concern for me than the experimental evaluation of the two techniques described above.
> > >
> >
> > We have previously attempted multi-round synthesis but encountered non-trivial challenges.
> >
> > Our method leverages Leandojo to interact with Lean and its traced ASTs to mutate the theorems. After the first round, the synthesized library becomes cumbersome, containing millions of theorems. It is hard to trace it using Leandojo.  Besides, the time required for multi-round synthesis substantially exceeds that of a single round due to the extensive number of seed theorems.
> >
> > To achieve successful multi-round synthesis, we need following techniques:
> >
> > - **Lighter and Faster interface**: As the number of theorems growing, the time-cost grows exponentially. A more lightweight and rapid interaction tool compared to Dojo could significantly reduce the time-cost.
> > - **Efficient Implementation for data-extraction**:  Mutation implementation relies on additional information provided by Lean (e.g., full_name of theorem, AST, and so on). Optimizing data-extraction process will be advantageous.
> > - **Metrics for quality-evaluation**: In multi-round synthesis, emphasis should be put on valuable variants while filtering the trivial mutations. Quality metrics (human-designed, model-generated or hybrid-ways.) may help refine the search process.
> >
> > > From what I understand, proofs are only modified by introducing a have statement that reverses the 1-step augmentation, and then the proof is the same as the original. Again, it would be nice to see additional innovation in this direction.
> > >
> >
> > Proofs in Alchemy-data are only modified by integrating a “have” with the original proofs.  Actually, there are many other ways to implement this (e.g., close the proof with ATP Tools or LLMs). We choose this pathway for two reasons: 1) It is a faster and more intuitional implementation compared with methods based on tools or models. 2) By constructing theorem variants established through a two-hop proof, we may facilitate improved learning capabilities for LLMs We will consider additional innovations as our future work.
> >
> > > It was unclear why each technique helped on unseen_premises split; could you give an intuition or an analysis of why it might help?
> > >
> >
> > For each technique in our method, we attempt to explain the rationale behind its effectiveness.
> >
> > - The CPT stage mainly help LLMs to be more adaptable to the traditional Best First Search, which utilizes cumulative logprob as heuristic for search.
> > - The inclusion of additional state-tactic pairs, focused on 'rw' and 'apply' tactics, aims to instruct the model on the specific utilization of the 'rw' and 'apply' tactics, respectively.
> >
> > Regarding the novel_premises split, as per the explanation in Leandojo [3], it indicates that the proof of a test theorem includes at least one premise usage that is not present in the training set. This prevents the model from simply memorizing training set to prove it. To prove a theorem containing novel premise, there are two pathways:
> >
> > - The model employs alternative premises that are adequate for proving the test theorem, thereby finding a distinct proof compared to the ground truth.
> > - The model develops a general reasoning ability for premise usage and endeavors to incorporate this new premise in the proof.
> >
> > Our method may potentially contribute to both aspects.
> >
> > ---
> >
> > [1] Dubey, Abhimanyu, et al. "The llama 3 herd of models." *arXiv preprint arXiv:2407.21783* (2024).
> >
> > [2] Guo, Daya, et al. "DeepSeek-Coder: When the Large Language Model Meets Programming--The Rise of Code Intelligence." *arXiv preprint arXiv:2401.14196* (2024).
> >
> > [3] Yang, Kaiyu, et al. "Leandojo: Theorem proving with retrieval-augmented language models." *Advances in Neural Information Processing Systems* 36 (2024).

---

> ### Author Response · Authors · 2024-11-25
> **Response to Reviewer FuWS**
>
> We want to thank you again for your time and patience in evaluating our work. Considering that the review time is soon coming to a close, we would greatly appreciate it if you have any further questions so that we can provide timely answers.

---

> > ### Comment · Reviewer_FuWS · 2024-11-26
> >
> > Thanks for all of the new experiments on continued pretraining and data overlap. They have addressed those concerns, so I'm raising my score from a 5 to a 6. Since the concerns about runtime, limited improvements, and a limited set of transformations still remain I would still consider this a borderline acceptance. Thank you again for your detailed responses!

---

> > > ### Author Response · Authors · 2024-11-27
> > > **Response to Reviewer FuWS**
> > >
> > > Thank you again for your time and efforts in evaluating our work. We deeply value your acknowledgement for our paper and are looking forward to refining our method in the future.

---

### Official Review · Reviewer_EDxu · 2024-11-04

**Soundness:** 4
**Presentation:** 3
**Contribution:** 2
**Rating:** 6
**Confidence:** 5

**Summary:**

This paper introduces a symbolic method called Alchemy to augment formal theorem proving data. Specifically, it mutates "the candidate theorem by replacing the corresponding term in the statement with its equivalent form or antecedent", which increases the number of theorem in mathlib4 from 110k to 6M. After continual pre-training and supervised fine-tuning with the generated data, it improves downstream performances (pass rate) on standard theorem proving benchmarks such as mathlib-test and miniF2F from 2.5% to 5%.

**Strengths:**

Some originality: This is a good and new attempt for augmenting a human-written library (although similar ideas have been applied for "pure symbolic and from scratch" methods for generating theorems such as INT, HTPS-Equations and AlphaGeometry)
Good quality: the paper is well written and all settings are of high relevance
Good clarity: the paper is presented in a clear manner. The experimental setting and results are presented in a standard way and easy to follow
Fair significance: The improvement on pass rate on mathlib-test and miniF2F is consistent, with almost all differences being positive compared with the baseline.

**Weaknesses:**

1. Poor improvement: although the improvement on pass rate is consistent, it's very limited: ranging from 0.62% to 4.7% on mathlib and only 2.47% on miniF2F (34.01% to 36.48%). This is pretty marginal in terms of improvement.
2. Narrow application possibility: the approach highly replies on a library of existing equivalence (or implying) theorems and their usage in proofs of other theorems.

**Questions:**

How do you explain a Conversion Ratio of only 37% while the idea seems to work with a theoretical guarantee (i.e. 100%)?
Do you think a framework like Alchemy is the correct way to significantly improve NTP to face challenging problems such as IMO problems?

---

> ### Author Response · Authors · 2024-11-20
> **Response to Reviewer EDxu**
>
> We would like to express our sincere gratitude to you for your time and effort in evaluating our work.
>
> > Poor improvement: although the improvement on pass rate is consistent, it's very limited: ranging from 0.62% to 4.7% on mathlib and only 2.47% on miniF2F (34.01% to 36.48%). This is pretty marginal in terms of improvement.
> >
>
> We have deliberated on the factors contributing to the relatively modest improvements and discussed potential refinements aimed at enhancing the performance of our method in the general response.
>
> > Narrow application possibility: the approach highly replies on a library of existing equivalence (or implying) theorems and their usage in proofs of other theorems.
> >
>
> Our symbolic mutation technique indeed relies on a formal library that comprises equality or implication rules and constructs new proofs by leveraging these theorems in conjunction with original proofs.  While this method necessitates certain prerequisites, we view its development as a valuable step towards exploring free-form theorem-synthesis methods within the symbolic space.
>
> > How do you explain a Conversion Ratio of only 37% while the idea seems to work with a theoretical guarantee (i.e. 100%)?
> >
>
> We have explained the reason behind the non-100% conversion ratio in the general response.
>
> > Do you think a framework like Alchemy is the correct way to significantly improve NTP to face challenging problems such as IMO problems?
> >
>
> As an exploration on data-synthesis in symbolic space, Alchemy has shown promising results on enhancing NTP. We assume Alchemy-like methods may indeed offer valuable assistance in tackling challenging problem sets like IMO problems.
>
> 1. Such methods, following the general spirit of AlphaGeometry [1], engage in random wandering within the symbolic space and synthesize new knowledge upon a well-designed symbolic framework. They may lay the groundwork for an AlphaGeo-style victory in Lean.
> 2. In practice, Alchemy-like method can be combined with existing NTP techniques.
>     1. It may serve as a statement-augmenter for autoformalized statements or theorem-augmenter before retraining for each round of expert iteration.
>     2. It can be used to augmenting existing knowledge base (available useful premises), which may be beneficial for Retrieval Augment Generation (RAG).
> 3. Transitioning from single-round mutations to multi-round mutations could potentially lead to the synthesis of exceedingly intricate and challenging theorems.
>
> ---
>
> [1] Trinh, Trieu H., et al. "Solving olympiad geometry without human demonstrations." *Nature* 625.7995 (2024): 476-482.

---

> > ### Comment · Reviewer_EDxu · 2024-11-26
> >
> > Thank you for your efforts of clarification. Very helpful! I still think this is a paper that can be accepted but only weakly because of the narrow application possibility and marginal improvements, which is rooted in the idea so cannot be easily changed. I'll maintain my score.

---

> > > ### Author Response · Authors · 2024-11-26
> > > **Response to Reviewer EDxu**
> > >
> > > Thank you again for your time in evaluating our work. We appreciate your recognition of the value of our work and look forward to further refining our methods in the future.

---

### Author Response · Authors · 2024-11-20
**General Response to the Shared Concerns or Questions**

We sincerely thank all reviewers for their valuable feedback and constructive comments in the reviewing process. We notice that some reviewers have similar concerns or questions.

1.  Poor improvement (**Reviewer EDxu, Reviewer BkaD**)
2.  Synthesis Cost (**Reviewer FuWS, Reviewer BkaD**)
3.  Non-100% Conversion Ratio (**Reviewer EDxu, Reviewer NSyk**)
4.  Data-Contamination (**Reviewer FuWS,** **Reviewer BkaD**)
5.  Data-Diversity (**Reviewer BkaD, Reviewer NSyk**)

We have carefully considered them and addressed them comprehensively below.

### 1. Poor Improvement

**Reviewer EDxu** and **Reviewer BkaD** point out that the improvements achieved by our method may be limited. **Reviewer BkaD** also compares the improvement achieved by our method versus the improvement of Deepseek Prover or InternLM Prover.

> **Reviewer EDxu:**
>
>
> Poor improvement: although the improvement on pass rate is consistent, it's very limited, ranging from 0.62% to 4.7% on mathlib and only 2.47% on miniF2F (34.01% to 36.48%). This is pretty marginal in terms of improvement.
>

> **Reviewer BkaD:**
>
>
> Marginal Gains in Benchmark Performance: Despite generating millions of new theorems, the gains in miniF2F accuracy are limited to 2.5%, notably lower than the >60% accuracy achieved by SOTA models such as DeepSeekProver and InternLM Prover. This modest improvement raises questions regarding the utility and quality of the synthetic theorems for real-world theorem-proving tasks.
>

We will explain the reason behind it and discuss the prevalent synthesis methods in Deepseek Prover or InternlmStepProver.

The limited improvements of Alchemy achieved in competition-level benchmarks might be attributed to the discrepancy between our synthesized data and competition-level theorems.   At the theorem level, our synthesized data is derived from fundamental theorems in Mathlib, which differ substantially from competition-level theorems. At the state-tactic level, as detailed in **Appendix E.2,** synthesized additional tactics of our algorithm are centered on basic tactics (rw and apply), rather than the advanced tactics (linarith, ring, omega, etc.) that are important for proving miniF2F-style theorems. We hypothesize that electing domain-similar seed theorems and focusing on synthesizing advanced tactics could enhance performance on miniF2F-like benchmarks.

The significant performance gains achieved by DeepseekProver [1] and InternLM Stepprover [2] primarily stem from expert iteration on a large set of competition-level statements that align with the downstream task (miniF2F). While these works have provided valuable insights and advanced the research of NTP, these methods face some limitations:

- They require extensive manual effort for collecting natural language problems and substantial computational resources (GPU-intensive) for formalization and proof generation.
- The distribution of formalized theorems is inherently constrained by the pool of human-collected natural language questions, creating limited new knowledge.

In contrast, constructing theorems in symbolic space offers a more direct pathway for generating new knowledge, eliminating the need for intermediate translation. This approach is also more scalable, leveraging cost-effective CPU resources. Our work explores this challenging yet unexplored direction, demonstrating its potential through improvements in both in-distribution and out-of-distribution benchmarks.

---

[1] Xin, Huajian, et al. "DeepSeek-Prover-V1. 5: Harnessing Proof Assistant Feedback for Reinforcement Learning and Monte-Carlo Tree Search." *arXiv preprint arXiv:2408.08152* (2024).
[2] Wu, Zijian, et al. "InternLM2. 5-StepProver: Advancing Automated Theorem Proving via Expert Iteration on Large-Scale LEAN Problems." *arXiv preprint arXiv:2410.15700* (2024).

---

> ### Author Response · Authors · 2024-11-20
> **General-Response-2**
>
> ### 2. High Synthesis Cost
>
> > **Reviewer FuWS**
> >
> >
> > The computational cost is very high; it takes 14 days for the rw operation on 512 CPU nodes. To make the authors' method more practical, it would have been nice to see some innovation that makes the extraction faster (either at the algorithmic level or the implementation level).
> >
>
> > **Reviewer BkaD**
> Computational Cost: The process of generating and verifying theorems is highly resource intensive. The implementation reports substantial computational overhead, with 14 days on 4,096 CPU cores for rw mutations and 7 days on 2,048 cores for apply mutations, potentially limiting the accessibility and scalability of Alchemy in practice.
> >
> >
> > Given the computational demands, are there potential optimizations in the synthesis process to reduce the time and resources required for theorem mutation?
> >
>
> **Reviewer FuWS** and **Reviewer BkaD** show their concerns about the huge cost of our synthesizing algorithm and expect some possible optimizations.
>
> ### Reason for the huge cost
>
> As detailed in **Section 4.1 and Appendix C.2,** the primary computational bottleneck stems from Lean interaction time.
>
> We choose Leandojo [1] as the tool to interact with Lean (run_tac API). The dojo version we used during the development of *Alchemy* is memory-intensive (requiring substantial memory usage and intensive IO), which hinders the implementation of multiprocessing. Besides, the initialization of the dojo is very slow (Several minutes for a dojo env).
>
> Due to the drawbacks of the dojo, we just split the target theorems into groups and send them to hundreds of CPU nodes. Nested for loops run on each node (for each target theorem t in this group, for each possible tactic instruction i, run_tac(t, i)).  This is a relatively slow but steady implementation on our existing hardware, compared to the multi-thread version (multi-dojo env for each node).
>
> ### Possible speedup methods
>
> The possible speedup methods are listed below:
>
> 1. **Leverage updated Leandojo features** Several updates about Leandojo may help decrease the cost. It significantly improves initialization speed when interacting with Lean4 after the 2.0.0 version and adds support for local and Remote Repositories after the 2.1.0 version [2].
> 2. **Develop a fast and light interface.**
>     - The Lean repl [3] has its advantages over Dojo.  It is lighter than Leandojo and friendly for multi-processing. Some Python wrappers [4, 5] for it are available, which may serve as bases for further development.
>     - However, the Lean repl also has its limitations. It requires a higher latency to extract the information.
>     - Based on the above discussion, we assume that it is promising to develop a fast interface for Lean based on Lean repl, which will not only speed up our algorithm a lot but also contribute to the research of Tree Search and Reinforcement Learning in NTP [6, 7, 8].
> 3. **Narrow search space** We can implement some heuristics or learn a model to narrow the search beam of possibly invocable theorems and help to avoid unnecessary operations.
> 4. **Scale the computing units (Trivial one)** It is much cheaper to extend the CPU than the GPU. Getting more CPU is the easiest way to lower the time cost.
>
> ---
>
> [1] Yang, Kaiyu, et al. "Leandojo: Theorem proving with retrieval-augmented language models." *Advances in Neural Information Processing Systems* 36 (2024).
>
> [2] https://github.com/lean-dojo/LeanDojo/releases?page=1
>
> [3] [leanprover-community/repl: A simple REPL for Lean 4, returning information about errors and sorries.](https://github.com/leanprover-community/repl)
>
> [4] [zhangir-azerbayev/repl: A simple REPL for Lean 4, returning information about errors and sorries.](https://github.com/zhangir-azerbayev/repl)
>
> [5] [cmu-l3/minictx-eval: Neural theorem proving evaluation via the Lean REPL](https://github.com/cmu-l3/minictx-eval)
>
> [6] Lample, Guillaume, et al. "Hypertree proof search for neural theorem proving." *Advances in neural information processing systems* 35 (2022): 26337-26349.
>
> [7] Xin, Huajian, et al. "DeepSeek-Prover-V1. 5: Harnessing Proof Assistant Feedback for Reinforcement Learning and Monte-Carlo Tree Search." *arXiv preprint arXiv:2408.08152* (2024).
>
> [8] [ABEL: Sample Efficient Online Reinforcement
> Learning for Neural Theorem Proving](https://openreview.net/pdf?id=kk3mSjVCUO)

---

> ### Author Response · Authors · 2024-11-20
> **General-response-3**
>
> ### 3. Non-100% Conversion Ratio
>
> **Reviewer EDxu** and **Reviewer NSyk** have questions about the non-100% conversion ratio from stage one to stage two.
>
> > **Reviewer EDxu**
> >
> >
> > How do you explain a Conversion Ratio of only 37% while the idea seems to work with a theoretical guarantee (i.e. 100%)?
> >
>
> > **Reviewer NSyk**
> >
> >
> > Why many generated theorems not pass the Lean prover? Since the generation process is based on symbolic replacement, I suppose most of the theorem should pass the prover.
> >
>
> We will recap the exact behavior of each stage and explain the reason why the conversion ratio is not equal to 100%.
>
> **As discussed in Appendix C**, our implementation consists of two stages.
>
> - Stage One Find invocable theorems for each target theorem by running tactics. Each invocable theorem is stored as a triplet (initial proof state, next proof state, tactic) as in **Fig 5.**
> - Stage Two We construct the mutated hypothesis or conclusion by parsing the next proof state and do symbolic replacement with the help of AST. Then, we build the new proof by integrating a “have” lemma with the original proof.
>
> Indeed, synthesizing theorems in symbolic space works with a theoretical guarantee when the symbolic system is robust and well-designed.  However, implementing the symbolic replacement is a non-trivial problem, which transforms codes in a pretty-printed proof state into raw lean code.
>
> Our implementation of symbolic replacement involves conducting various string manipulations and parsing the ASTs for localization. Although conceptually straightforward, this method grapples with intricate scenarios like meta-variables, coercions, and other complexities.
>
> For example, when replacing the old hypothesis of the target theorem with subgoals introduced by the invocable theorem for “apply”, navigating the relationship between metavariables [1] (e.g.,?a,?u.1338287) in the next proof state may be complex.  Analyzing these relationships and assigning valid values to fill the gaps accurately poses a significant challenge, especially when conflicts arise in variable naming. Our conflict-detection and renaming mechanism [2] may falter in handling such intricate scenarios.
>
> The complex metavariables cases account for a large ratio of the unpassed theorems, which is hard to tackle using several rules. We speculate that leveraging Large Language Models (LLMs) to fill these holes could offer a potential solution.
>
> Despite these hurdles, our current implementation has successfully synthesized over three million theorems, augmenting the theorem-proving capacity of LLMs. Improving our implementation will further increase the conversion ratio, which requires a meticulous examination of the Lean parser and elaborator.
>
> ---
>
> [1] [MetaM - Metaprogramming in Lean 4](https://leanprover-community.github.io/lean4-metaprogramming-book/main/04_metam.html)
>
> [2] [Mathlib naming conventions](https://leanprover-community.github.io/contribute/naming.html)

---

> > ### Author Response · Authors · 2024-11-24
> > **General Response**
> >
> > We want to express our sincere gratitude to all reviewers again. If there exists any lingering questions that remain unanswered in our response to you, we are eager to provide further details and engage in discussion with you.

---

> ### Author Response · Authors · 2024-11-20
> **General-Response-4-1**
>
> ### 4. Data Contamination
>
> **Reviewer FuWS** and **Reviewer BkaD** express similar concerns about the data-contamination problems.
>
> > **Reviewer FuWS**
> >
> >
> > The LeanDojo benchmark consists of theorems from Mathlib. Therefore, there is potential train-test overlap in at least two places.
> >
> > - (i) First, the continued pretraining dataset, if it includes theorems from the LeanDojo test set (or premises used in the novel_premises split). How was train-test overlap prevented for continued pretraining? I wasn't able to find details on exactly what was done for continued pretraining, so it would be great to clarify this.
> > - (ii) Second, the rewrites and applies may use premises that are "novel" in the novel_premises split. How do you ensure that these are not used in the data augmentation process?
>
> > **Reviewer BkaD**
> How do you avoid the data contamination problem in the evaluation/generation phase?
> >
>
> We take the data contamination problem seriously. We will show as many details about our work on this topic as possible.
>
> ### The format of our synthesized data
>
> We synthesize data **with the whole mathlib dataset and do deduplication as in the following sections**. The synthesized data are stored in jsonl format. Each line is as follows.
>
> ```json
> {
> 	"file_name": the name of the lean file in mathlib,
> 	"original_text": the content of the file before writing variants back,
> 	"text": the content of the file with variants
> 	# we store the line number of each mutated
> 	# variant with its original theorem name as key, [line_start, line_end]
> 	"loc": {
> 			“theorem_name_1": [[20, 24], [25, 29]....],
> 			“theorem_name_2": [[122, 127], [128, 133]....],
> 			“theorem_name_3": [[222, 227], [228, 233]....]
> 			...
> 	},
> 	# valid_loc has the same format as loc. But it only stores the variants
> 	# that passes the check of theorem prover (after Stage Two)
> 	"valid loc": {...},
> 	"meta": meta information (url, commit)
> }
> ```
>
> With the location and original name of each variant recorded, we are capable of conducting thorough data de-contamination.
>
> ### Details of Continual Pre-Training
>
> We conduct continual pre-training at the theorem level.  An example of our training data is shown in **Fig 10.** Besides, as shown in **Fig 6,** the number of variants of different target theorems varies a lot. To mitigate the risk of biased learning due to this imbalance, we reduce the number of variants for each original theorem to adhere to a predefined maximum threshold.
>
> ### De-contamination
>
> Our training data for CPT and SFT are composed of two parts:
>
> - **Mathlib-train**: Theorems (State-Tactics) in the **training set** of respective splits (random, novel_premises)
> - **Synthetic Data:** Mutated Theorems (Additional Synthesized State-Tactics)
>
> We try our best to avoid the train-test overlap:
>
> 1. Each model evaluated on different splits (random, novel_premises) is trained on distinct data. That’s to say, for a single line in **Table 3,** we need to train two models.
>     1. **Mathlib-train** is the corresponding training set of the specific split
>     2. **Synthetic Data** comprises unique subsets of our synthesized data achieved by excluding variants of theorems and their associated state-tactics pairs present in the test split.
> 2. Our training datasets strictly exclude theorems and variants from the test split.
>     1. **CPT Dataset**:  We eliminate all theorems and their synthesized variants present in the test split from the CPT dataset by matching theorem names.
>     2. **SFT Dataset:** State-tactic pairs traced from the theorems in the test split and their corresponding synthesized variants are removed from the SFT dataset.
> 3. As for the novel_premises split, according to the explanation in Leandojo [1], it indicates that the proof of a test theorem includes at least one premise usage that is not present in the training set. In response to **Reviewer FuWS**'s concerns regarding the effectiveness of the novel_premises benchmark and potential train-test overlaps with the data construction, we conduct a post-analysis. The whole procedure is as follows:
>     1. We identify the novel premises by comparing the used premises in the training set and test set of the Leandojo Benchmark leveraging annotations provided by Leandojo [1].
>     2. We parse the introduced “have” lemma in the CPT dataset and parse the additional state-tactic pairs in the SFT dataset that contain the novel premises (via simple regex matchings).
>     3. We undertake additional training to rectify any issues with the experimental setup by removing such overlaps and retraining the model.
>
> ### Novel-Premise overlap
>
> We show the overlap ratios (num_containing_premise/total_num) in the table below:
>
> |  Data Type | rw | apply | total |
> | --- | --- | --- | --- |
> | CPT | 1.9% | 0.3% | 1.1% |
> | SFT |  1.2% | 0.6% |  1% |
>
> We observed that the overlap ratio is relatively low, suggesting that its impact on improvement might be marginal.

---

> ### Author Response · Authors · 2024-11-20
> **General-Response-4-2**
>
> ### Retraining Experiments
>
> We remove the overlap data in our dataset and retrain the Llama-3-8b. The cleaned CPT data is now referred to as cpt-clean, while the cleaned SFT data is labeled as sft-clean. Their respective original training datasets, "Mathlib-train + rw + apply," are denoted as cpt-old and sft-old in our framework.
>
> - CPT-ablation (all experiments with mathlib-tain sft)
>
> | setting | novel_premises |
> | --- | --- |
> | mathlib-train-cpt | 39.54% |
> | cpt-old  | 42.19% |
> | cpt-clean  | 41.90% (-0.29%) |
> - SFT-ablation (all experiments without cpt)
>
> | setting | novel_premises |
> | --- | --- |
> | mathlib-train-sft | 38.52% |
> | sft-old  | 41.95% |
> | sft-clean | 41.17% (-0.78%) |
> - CPT + SFT-ablation
>
> | setting  | novel_premises |
> | --- | --- |
> | cpt-old + sft-old | 43.22% |
> | cpt-clean + sft-clean | 43.16% (-0.06%) |
>
> The experimental results show that the overlap contributes a little to our improvement.
>
> ---
>
> [1] Yang, Kaiyu, et al. "Leandojo: Theorem proving with retrieval-augmented language models." *Advances in Neural Information Processing Systems* 36 (2024).

---

> ### Author Response · Authors · 2024-11-20
> **General Response-5**
>
> ### 5. Data Diversity Issue
>
> **Reviewer BkaD** and **Reviewer NSyk** show their concern about the lack of metrics for evaluating the diversity of synthesized theorems.
>
> > **Reviewer BkaD**
> Lack of Quality Metrics for Synthetic Theorems: Although Alchemy generates a large corpus, there is limited analysis of the quality or mathematical significance of the produced theorems. Without metrics or evaluation methods beyond correctness by construction, it is challenging to assess whether the synthetic theorems provide meaningful, diverse training examples.
> >
> >
> >
> > Given the modest improvement in miniF2F accuracy, are there metrics or quality checks available to assess the mathematical value or diversity of the generated theorems beyond correctness?
> >
>
> > **Reviewer NSyk**
> >
> >
> > The method seems unable to generate diverse theorem data. It mainly expands existing theorem by combining other theorems. The diversity problem may result in a lower improvement on the harder benchmark miniF2F. I guess the generated theorem can be very different from the original theorem if it has a deep variant proof tree. Authors may show the depth statistics of the generated theorem or other statistics to verify the diversity of the generated theorem.
> >
>
> In our methodology, mutations are applied to the statements of each theorem, capturing the essence of the theorems. Synthesized statements that successfully pass the Lean can be considered meaningful theorems to a certain degree. Additionally, our approach involves merging two existing proof trees from the Lean Mathematical Library, ensuring the significance of the generated theorems. As illustrated in Figure 1, a statement can undergo mutation to produce meaningful variants with mathematical meanings distinct from the original theorem.
>
> To give deeper information about the diversity of our generated statements, we compute the Rouge score [1], a metric used in automatic summary generation tasks to evaluate the text similarity between the reference summary and generated summary. Specifically, with a reference sentence *ref* and a generated sentence *gen,* it computes the similarity between them.
>
> We define below metrics to evaluate the diversity of generated theorems:
>
> 1. intra-diversity: A metric that evaluates how different the mutated theorems are compared to their original theorems and shows the effectiveness of our mutations. We select the original theorem as *ref* and its variants as *gen.* For each original theorem, we compute an average Rouge score. The returning score is the average of scores of all original theorems.
> 2. inter-diversity: A metric that evaluates the diversity of all synthesized variants. We adopt a bootstrap-like method. For each variant, we randomly sample twenty variants from the dataset as *refs* and compute the average score. The returning score is the average of scores of all variants.
>
> For all these metrics, the lower, the better. The scores are listed in the table below: (Rouge-L)
>
> | metric | rw | apply | Avg | Original |
> | --- | --- | --- | --- | --- |
> | intra-diversity | 0.56 | 0.48 | 0.52 | - |
> | inter-diversity | - | - | 0.167 | 0.164 |
>
> The intra-diversity score of 0.52 indicates that our synthesized statements differ from the original theorems, demonstrating the effectiveness of our mutation process. Furthermore, we have noticed that the "apply" method outperforms the "rw" method in terms of mutation.
>
> With an inter-diversity score of 0.167, we note a high level of diversity among the synthesized theorems. This score is nearly matching the original inter-diversity score, which means our method does not lower the diversity of original data.
>
> In summary, our mutation methodology proves effective in generating a range of mutated theorems. Besides, as **Reviewer** **NSyk** said, synthesizing theorems in multi-round and generating deeper proof trees may further improve the diversity of generated theorems.
>
> ---
>
> [1] Lin, Chin-Yew. ROUGE: a Package for Automatic Evaluation of Summaries. In Proceedings of the Workshop on Text Summarization Branches Out (WAS 2004), Barcelona, Spain, July 25 - 26, 2004.

---

### Meta-Review · Area_Chair_unQU · 2024-12-25

**Metareview:**

This paper concerns data augmentation for theorem proving in Lean through symbolic rewriting of hypotheseses and proofs. Evaluations performed on Mathlib datasets show that, with data augumentation, further pretraining and finetuning improves the performance by 2.69% on the random LeanDojo test split and 4.22% on the novel_premises split. Augmenting training data for theorem proving through rewrites and applies is a novel contribution. Given the augmentation only performs one step rewriting, leaving the augumented theorems and proofs nearly identical to the original ones, there is a concern that whether the fine-tuned model will generalize. Other concerns are possible train-test overlap and data contamination. New results shared during the rebuttal partially addresses some of these concerns (e.g., train-test overlap). Overall, this paper makes valuable contributions with relatively marginal improvement by data augmentation for neural theorem proving.

**Additional Comments On Reviewer Discussion:**

There were active discussions between reviewers and authors during the rebuttal. Main concerns raised by reviewers are train-test overlap and data contamination. The authors conducted a duplication analysis and found a small fraction of overlap, and with de-duplication applied, new results show that there is a slight performance drop but the improvement is still consistent.

---

### Decision · Program_Chairs · 2025-01-22

Accept (Poster)